# Off-Policy Evaluation and Learning from Logged Bandit Feedback: Error Reduction via Surrogate Policy

**Yuan Xie**[*]
Indiana University Bloomington
xieyuan@umail.iu.edu

**Boyi Liu**[*]
Northwestern University
boyiliu2018@u.northwestern.edu

**Qiang Liu**
The University of Texas at Austin
lqiang@cs.utexas.edu

**Zhaoran Wang**
Northwestern University
zhaoranwang@gmail.com

**Yuan Zhou**
Indiana University Bloomington
yzhoucs@iu.edu

**Jian Peng**
University of Illinois at Urbana-Champaign
jianpeng@illinois.edu

## Abstract

When learning from a batch of logged bandit feedback, the discrepancy between the policy to be learned and the off-policy training data imposes statistical and computational challenges. Unlike classical supervised learning and online learning settings, in batch contextual bandit learning, one only has access to a collection of logged feedback from the actions taken by a historical policy, and expect to learn a policy that takes good actions in possibly unseen contexts. Such a batch learning setting is ubiquitous in online and interactive systems, such as ad platforms and recommendation systems. Existing approaches based on inverse propensity weights, such as Inverse Propensity Scoring (IPS) and Policy Optimizer for Exponential Models (POEM), enjoy unbiasedness but often suffer from large mean squared error. In this work, we introduce a new approach named Maximum Likelihood Inverse Propensity Scoring (MLIPS) for batch learning from logged bandit feedback. Instead of using the given historical policy as the proposal in inverse propensity weights, we estimate a maximum likelihood surrogate policy based on the logged action-context pairs, and then use this surrogate policy as the proposal. We prove that MLIPS is asymptotically unbiased, and moreover, has a smaller nonasymptotic mean squared error than IPS. Such an error reduction phenomenon is somewhat surprising as the estimated surrogate policy is less accurate than the given historical policy. Results on multi-label classification problems and a large-scale ad placement dataset demonstrate the empirical effectiveness of MLIPS. Furthermore, the proposed surrogate policy technique is complementary to existing error reduction techniques, and when combined, is able to consistently boost the performance of several widely used approaches.

## 1 Introduction

While maintaining online and interactive systems for information retrieval, news recommendation, ad placement, and e-commerce, we collect large batches of logs during testing phases and past deployment periods (Li et al., 2010). Such logs contain rich information that can be used for many purposes. For example, the logs of an e-commerce system record the decisions (or actions) on which ads or recommendations are displayed to a user given a context, and the corresponding feedback (or reward), including whether the user clicks on any of the displayed items and whether a purchase

---

[*]equal contribution, the order of the two co-first authors is determined by a coin toss

occurs afterwards. Although the logs are collected when the online system is deployed with a historical algorithm (or policy), they are informative for future improvements and design of the system. One application of such logs is to estimate the performance of new policies, also known as off-policy evaluation (Dudík et al., 2014; Li et al., 2011; 2014). Furthermore, the use of logs allows for an accurate and more efficient way to test and optimize new policies without expensive trial-and-error cycles in traditional A/B tests, which often take weeks.

Since the logged feedback only provides partial information, policy evaluation and optimization using logs are often formulated as batch contextual bandit learning. Different from the online learning setting, batch contextual bandit learning is statistically more challenging because the collected logs are off-policy, that is, they are generated by a historical policy that differs from the current policy we intend to evaluate or optimize. To bridge this discrepancy, Inverse Propensity Scoring (IPS) has been introduced to construct an unbiased counterfactual estimator of policy performance using off-policy data (Bottou et al., 2013). However, such an estimator tends to have a large mean squared error when the current policy and the historical policy are very different. To reduce the mean squared error, a number of regularization and control variate approaches have been introduced (e.g., Dudík et al. (2014); Hirano et al. (2003); Ionides (2008); Li et al. (2015)). Notably, Swaminathan & Joachims (2015a) proposed Policy Optimizer for Exponential Models (POEM), which introduces a mean squared error regularizer based on the counterfactual risk minimization principle. Empirical evaluations also suggested the effectiveness of these approaches on policy evaluation and optimization.

In this paper, we introduce a new yet simple parametric approach for mean squared error reduction in batch contextual bandit learning. Instead of using the given historical policy, we use linear models or neural network models to estimate a maximum likelihood surrogate policy using the logged action-context pairs. Then we use such a surrogate policy as the proposal distribution to obtain the inverse propensity weights in the off-policy estimator, as if the logs are generated by this surrogate policy. This idea dates back to the "estimated sampler" technique in statistics (Delyon & Portier, 2016; Henmi et al., 2007). We provide theoretical justification for this approach, which is named Maximum Likelihood Inverse Propensity Scoring (MLIPS). In particular, we show that the proposed MLIPS estimator is asymptotically unbiased, and moreover, has a smaller nonasymptotic mean squared error than the IPS estimator. Such an error reduction effect is counterintuitive as we replace the known historical policy with a less accurate estimated surrogate policy, which is supposed to increase the mean squared error. We evaluate the MLIPS estimator on several multi-label classification problems and a large-scale ad placement dataset to demonstrate its empirical effectiveness on mean squared error reduction for policy evaluation and optimization. Furthermore, we also combine the MLIPS estimator with several existing approaches. We find that our surrogate policy technique is complementary to existing mean squared error reduction techniques and achieves consistently improved performance.

## 2 OFF-POLICY EVALUATION AND LEARNING FROM LOGGED BANDIT FEEDBACK

In this section, we introduce the background on off-policy evaluation and learning from logged bandit feedback. In particular, we focus on IPS and present several existing mean squared error reduction and regularization approaches.

### 2.1 INVERSE PROPENSITY SCORING

The value function used to evaluate a target policy $\pi$ is defined as

$$V = \mathbb{E}_\pi[r] = \mathbb{E}_{x \sim \nu} \mathbb{E}_{a \sim \pi(\cdot \mid x)} \mathbb{E}_{r \sim D(\cdot \mid a,x)}[r]. \tag{2.1}$$

Here $x$ is the context drawn from the context distribution $\lambda$, which is independent of $\pi$, $a$ is the action taken according to $\pi$ given context $x$, and $r$ is the reward (or feedback) received.

In the off-policy setting, we are not able to sample different actions from the target policy $\pi$ and obtain reward, which is expensive and time-consuming in practice. Instead, we collect samples from a logging policy $\mu$, that is, $a \sim \mu(\cdot \mid x)$, and expect to use the logs to evaluate a target policy and further perform policy optimization.

To bridge the discrepancy between $\pi$ and $\mu$, the IPS estimator, which is known as the importance sampling estimator, reweights each logged action-context pair as follows,

$$\widetilde{V}_{\text{IPS}} = \frac{1}{n} \sum_{i=1}^{n} \rho(x_i, a_i) \cdot r_i = \frac{1}{n} \sum_{i=1}^{n} \frac{\pi(a_i \,|\, x_i)}{\mu(a_i \,|\, x_i)} \cdot r_i. \tag{2.2}$$

In the sequel, we use $\widetilde{V}$ to denote $\widetilde{V}_{\text{IPS}}$ for notational simplicity.

It is worth noting that the IPS estimator is an unbiased estimator of $V$, and thus has been widely adopted as an objective function for optimizing the target policy $\pi$. However, $\rho(x_i, a_i) = \pi(a_i \,|\, x_i)/\mu(a_i \,|\, x_i)$ in (2.2) may induce a large mean squared error, especially when the logging policy $\mu$ is very different from the target policy $\pi$. As a result, the IPS estimator often leads to unsatisfactory policy optimization and poor generalization performance.

## 2.2 Mean Squared Error Reduction and Regularization

To reduce the mean squared error of IPS, several thresholding approaches have been proposed and studied in the literature (e.g., Bottou et al. (2013); Cortes et al. (2010); Ionides (2008); Strehl et al. (2010)). One straightforward approach is Propensity Weight Capping (Ionides, 2008), which takes the form

$$\widetilde{V}_{\text{IPS}}^{(M)} = \frac{1}{n} \sum_{i=1}^{n} \min\left\{ M, \frac{\pi(a_i \,|\, x_i)}{\mu(a_i \,|\, x_i)} \right\} \cdot r_i, \tag{2.3}$$

where $M > 1$ is the threshold parameter. Although thresholding reduces the mean squared error when $M$ is small, it introduces a large bias.

More recently, Swaminathan & Joachims (2015a) proposed the POEM algorithm for batch learning from bandit feedback. POEM is motivated by the structural risk minimization principle and a generalization bound that characterizes the mean squared error of $\widetilde{V}_{\text{IPS}}^{(M)}$ based on an empirical Bernstein argument. The POEM algorithm jointly optimizes $\widetilde{V}_{\text{IPS}}^{(M)}$ and its empirical standard deviation, which is also known as counterfactual risk minimization.

Let $u_{\pi}^{i} = \min\{M, \pi(a_i \,|\, x_i)/\mu(a_i \,|\, x_i)\} \cdot r_i$, $\bar{u}_{\pi} = \sum_{i=1}^{n} u_{\pi}^{i}/n$, and $\widehat{\text{Var}}_{\pi}(\widetilde{V}_{\text{IPS}}^{(M)}) = (\sum_{i=1}^{n}(u_{\pi}^{i} - \bar{u}_{\pi})^2)/(n-1)$, the POEM algorithm searches for a policy $\pi$ that maximizes

$$\widetilde{V}_{\text{IPS}}^{(M)} - \lambda \sqrt{\widehat{\text{Var}}_{\pi}(\widetilde{V}_{\text{IPS}}^{(M)})/n}. \tag{2.4}$$

Here $M > 1$ and $\lambda \geq 0$ are the thresholding and regularization parameters, respectively. Also note that when $\lambda = 0$, the POEM objective in (2.4) reduces to (2.3).

Motivated by the propensity overfitting problem of IPS and POEM, Swaminathan & Joachims (2015b) proposed to use control variates, which lead to the following self-normalized estimator

$$\widetilde{V}_{\text{SN}} = \frac{1/n \cdot \sum_{i=1}^{n} \pi(a_i \,|\, x_i)/\mu(a_i \,|\, x_i) \cdot r_i}{1/n \cdot \sum_{i=1}^{n} \pi(a_i \,|\, x_i)/\mu(a_i \,|\, x_i)}. \tag{2.5}$$

Note that $\widetilde{V}_{\text{SN}}$ is shifted by a constant $C$ if each $r_i$ is shifted to $r_i + C$. In other words, $\widetilde{V}_{\text{SN}}$ is equivariant (Hesterberg, 1995) and always lies within the range of $r_i$. In contrast, the IPS estimator does not have such properties. Self-Normalized POEM (Norm-POEM), a variant of POEM, was also proposed by Swaminathan & Joachims (2015b). In particular, Norm-POEM searches for a policy that maximizes the normalized estimator regularized by its empirical standard deviation

$$\sqrt{\widehat{\text{Var}}_{\pi}(\widetilde{V}_{\text{SN}})} = \sqrt{\frac{1/n \cdot \sum_{i=1}^{n}(r_i - \widetilde{V}_{\text{SN}})^2 \cdot \left(\pi(a_i \,|\, x_i)/\mu(a_i \,|\, x_i)\right)^2}{\left(1/n \cdot \sum_{i=1}^{n} \pi(a_i \,|\, x_i)/\mu(a_i \,|\, x_i)\right)^2}}. \tag{2.6}$$

In addition, other techniques, such as control variates and doubly-robust estimators Dudík et al. (2014) have also been applied to reduce the mean squared error of the off-policy estimator.

## 3  ERROR REDUCTION VIA SURROGATE POLICY

Instead of directly using the logging policy propensities, we propose to refit a surrogate policy and use it to obtain the inverse propensity weights in the IPS estimator defined in (2.2). We assume that the logging policy $\mu(a \mid x)$ is parameterized by $\beta \in \mathbb{R}^d$ and denote it by $\mu(a \mid x; \beta)$. In particular, we assume the logging policy with $\beta = \beta^*$ generates the logged action-context pairs. Although we have the access to $\mu(a \mid x; \beta^*)$, we choose to use the maximum likelihood estimator of $\beta^*$ based on $\{(a_i, x_i)\}_{i=1}^n$ as a surrogate policy parameter, which is denoted as $\widehat{\beta}$, to replace $\beta^*$ and obtain a more accurate estimator of the value function $V$ defined in (2.1). In particular, we define the MLIPS estimator of $V$ as

$$\widehat{V} = \frac{1}{n} \sum_{i=1}^n \frac{\pi(a_i \mid x_i)}{\mu(a_i \mid x_i; \widehat{\beta})} \cdot r_i, \tag{3.1}$$

where $\{(a_i, x_i)\}_{i=1}^n$ are the logged action-context pairs, which are independently sampled from $\mu(a \mid x; \beta^*)$.

Unlike the IPS estimator $\widetilde{V}$ in (2.2), the MLIPS estimator $\widehat{V}$ in (3.1) is biased in general. However, $\widehat{V}$ is asymptotically unbiased, and moreover, in the following subsection we show that $\widehat{V}$ achieves a smaller mean squared error than $\widetilde{V}$ when the logging distribution $\mu(a \mid x; \beta)$ is properly chosen.

### 3.1  THEORETICAL ANALYSIS

In this section, for the simplicity of analysis, we assume that the reward $r$ is determined by $a$ and $x$, that is, $r = r(a, x)$. For notational simplicity, in the subsequent analysis we use the following equivalent definitions of $\widetilde{V}$ and $\widehat{V}$ based on joint distributions,

$$\widetilde{V} = \frac{1}{n} \sum_{i=1}^n \frac{\pi(a_i, x_i)}{\mu(a_i, x_i; \beta^*)} \cdot r(a_i, x_i), \quad \widehat{V} = \frac{1}{n} \sum_{i=1}^n \frac{\pi(a_i, x_i)}{\mu(a_i, x_i; \widehat{\beta})} \cdot r(a_i, x_i),$$

since we have

$$\frac{\pi(a \mid x)}{\mu(a \mid x; \beta)} = \frac{\pi(a, x)/\lambda(x)}{\mu(a, x; \beta)/\lambda(x)} = \frac{\pi(a, x)}{\mu(a, x; \beta)}. \tag{3.2}$$

For the ease of notation, we denote $\mathbb{E}_{(a,x) \sim \mu(\cdot, \cdot; \beta^*)}[\,\cdot\,]$ by $\mathbb{E}[\,\cdot\,]$. Also, we denote by $\|\cdot\|$ the $\ell_2$-norm of a vector or the spectral norm of a matrix. For a function $f(\beta)$, we denote by $\partial f(\beta^*)/\partial \beta$ its derivative evaluated at $\beta = \beta^*$, which is $\partial f(\beta)/\partial \beta|_{\beta=\beta^*}$.

Before we lay out the main theory, we first introduce the following definitions.

**Definition 3.1** (Score Function and Fisher Information)**.**  The score function $S(a, x; \beta) \in \mathbb{R}^d$ and the Fisher Information matrix $I(\beta) \in \mathbb{R}^{d \times d}$ are defined as

$$S(a, x; \beta) = \frac{\partial \log \mu(a, x; \beta)}{\partial \beta}, \quad I(\beta) = -\mathbb{E}\left[\frac{\partial S(a, x; \beta)}{\partial \beta}\right]. \tag{3.3}$$

It is worth noting that the reformulation from $\mu(a \mid x; \beta)$ to $\mu(a, x; \beta)$ does not change the score function or the Fisher information matrix, since the marginal distribution of the context $x$ in (3.2), namely $\lambda(x)$, does not depend on $\beta$, and hence its partial derivative with respect to $\beta$ is zero.

**Definition 3.2** (Deviation Function)**.**  The difference between $\pi(a, x)/\mu(a, x; \beta) \cdot r(a, x)$ and the true value $V$ of the target policy, which is defined in (2.1), is denoted by

$$D_V(r, a, x; \beta) = \frac{\pi(a, x)}{\mu(a, x; \beta)} \cdot r(a, x) - V. \tag{3.4}$$

In the sequel, we introduce two assumptions and a condition on an event. The next assumption ensures that the Fisher information matrix of the joint logging distribution $\mu(a, x; \beta^*)$ is well behaved and the identifiability of the model is thus ensured.

**Assumption 3.3** (Nonsingular Fisher Information). For the joint distribution $\mu(\cdot, \cdot; \beta^*)$, we assume that its Fisher information matrix satisfies

$$\|I^{-1}(\beta^*)\| = \left\| \mathbb{E}\left[ \frac{\partial S(a, x; \beta^*)}{\partial \beta} \right] \right\|^{-1} = O(1). \tag{3.5}$$

The next assumption ensures the score and deviation functions are sufficiently smooth.

**Assumption 3.4** (Smoothness of Score and Deviation). For the deviation function $D_V(r, a, x; \beta)$ defined in (3.4), we assume that

$$\left\| \mathbb{E}\left[ \frac{\partial D_V(r, a, x; \beta^*)}{\partial \beta} \right] \right\| = O(\zeta_{\mathrm{DG}}), \quad \left\| \mathbb{E}\left[ \frac{\partial^2 D_V(r, a, x; \beta)}{\partial \beta^2} \right] \right\| = O(\zeta_{\mathrm{DH}}), \tag{3.6}$$

where the second equality holds for any $\beta$. Meanwhile, for the score function $S(a, x; \beta)$ defined in (3.3), we assume that for any $v \in \mathbb{R}^d$ such that $\|v\| = 1$, it holds that

$$\left\| \mathbb{E}\left[ \frac{\partial^2 S(a, x; \beta)}{\partial \beta^2}(v, v, \cdot) \right] \right\| = O(\zeta_{\mathrm{ST}}), \tag{3.7}$$

where $\partial^2 S(a, x; \beta)/\partial \beta^2$ is a tensor consisting of the third-order derivative of $\log \mu(a, x; \beta)$ with respect to $\beta$. Here $(\partial^2 S(a, x; \beta_0)/\partial \beta^2)(v_1, v_2, \cdot)$ is the bilinear map $\mathbb{R}^d \times \mathbb{R}^d \mapsto \mathbb{R}^d$ induced by the tensor, which maps $(v_1, v_2)$ to a vector in $\mathbb{R}^d$.

The Assumptions 3.3-3.4 are satisfied, under necessary regularity conditions, by many popular models such as the multinomial logistic regression model which will be illustrated in detail in §D. Before introducing the condition, we define several events to simplify the presentation. In the following, we first define an event on the estimation error of the maximum likelihood estimator $\widehat{\beta}$.

**Definition 3.5** (Maximum Likelihood Estimation Error). For the maximum likelihood estimator $\widehat{\beta}$ and the true parameter $\beta^*$, we define the event $\mathcal{E}_\beta(t_\beta)$ as

$$\mathcal{E}_\beta(t_\beta) = \left\{ \|\widehat{\beta} - \beta^*\| \leq t_\beta \right\}. \tag{3.8}$$

Next we define several events on the concentration behavior of the score function defined in (3.3) and the deviation function defined in (3.4) as well as their derivatives.

**Definition 3.6** (Concentration of Score). For the score function with true parameter $\beta^*$, we define the following events,

$$\mathcal{E}_{\mathrm{SG}}(t_{\mathrm{SG}}) = \left\{ \left\| \frac{1}{n} \sum_{i=1}^{n} S(a_i, x_i; \beta^*) - \mathbb{E}\left[ S(a, x; \beta^*) \right] \right\| \leq t_{\mathrm{SG}} \right\}, \tag{3.9}$$

$$\mathcal{E}_{\mathrm{SH}}(t_{\mathrm{SH}}) = \left\{ \left\| \frac{1}{n} \sum_{i=1}^{n} \frac{\partial S(a_i, x_i; \beta^*)}{\partial \beta} - \mathbb{E}\left[ \frac{\partial S(a, x; \beta^*)}{\partial \beta} \right] \right\| \leq t_{\mathrm{SH}} \right\}. \tag{3.10}$$

Also, we define

$$\mathcal{E}_{\mathrm{ST}}(t_{\mathrm{ST}}) = \left\{ \sup_{\beta, \|v\|=1} \left\| \frac{1}{n} \sum_{i=1}^{n} \frac{\partial^2 S(a_i, x_i; \beta)}{\partial \beta^2}(v, v, \cdot) - \mathbb{E}\left[ \frac{\partial^2 S(a, x; \beta)}{\partial \beta^2}(v, v, \cdot) \right] \right\| \leq t_{\mathrm{ST}} \right\},$$

where $(\partial^2 S(a, x; \beta)/\partial \beta^2)(v, v, \cdot)$ is defined in Assumption 3.4. Then we define the joint event $\mathcal{E}_{\mathrm{S}}(t_{\mathrm{S}}) = \mathcal{E}_{\mathrm{SG}}(t_{\mathrm{SG}}) \cap \mathcal{E}_{\mathrm{SH}}(t_{\mathrm{SH}}) \cap \mathcal{E}_{\mathrm{ST}}(t_{\mathrm{ST}})$, where $t_{\mathrm{S}} = (t_{\mathrm{SG}}, t_{\mathrm{SH}}, t_{\mathrm{ST}})$.

**Definition 3.7** (Concentration of Deviation). For deviation function with true parameter $\beta^*$, namely $D_V(r, a, x; \beta^*)$, we define

$$\mathcal{E}_{\mathrm{DG}}(t_{\mathrm{DG}}) = \left\{ \left\| \frac{1}{n} \sum_{i=1}^{n} \frac{\partial D_V(r_i, a_i, x_i; \beta^*)}{\partial \beta} - \mathbb{E}\left[ \frac{\partial D_V(r, a, x; \beta^*)}{\partial \beta} \right] \right\| \leq t_{\mathrm{DG}} \right\}. \tag{3.11}$$

Also, we write

$$\mathcal{E}_{\mathrm{DH}}(t_{\mathrm{DH}}) = \left\{ \sup_{\beta \in \mathbb{R}^d} \left\| \frac{1}{n} \sum_{i=1}^{n} \frac{\partial^2 D_V(r_i, a_i, x_i; \beta)}{\partial \beta^2} - \mathbb{E}\left[ \frac{\partial^2 D_V(r, a, x; \beta)}{\partial \beta^2} \right] \right\| \leq t_{\mathrm{DH}} \right\}.$$

Then we define the joint event $\mathcal{E}_{\mathrm{D}}(t_{\mathrm{D}}) = \mathcal{E}_{\mathrm{DG}}(t_{\mathrm{DG}}) \cap \mathcal{E}_{\mathrm{DH}}(t_{\mathrm{DH}})$, where $t_{\mathrm{D}} = (t_{\mathrm{DG}}, t_{\mathrm{DH}})$.

We now present a condition on the overall tail behavior of the probability of the events defined above.

**Condition 3.8** (Concentration Tail Behavior). Let $t^\star = (t_\beta, t_S, t_D)$ and
$$\mathcal{E}(t^\star) = \mathcal{E}_\beta(t_\beta) \cap \mathcal{E}_S(t_S) \cap \mathcal{E}_D(t_D), \tag{3.12}$$
it holds that $\mathbb{P}(\mathcal{E}(t^\star)) \geq 1 - f(t)$, where $t = \|t^\star\|_\infty$ for some decreasing tail function $f(t)$ such that $\int_0^\infty t^k \, df(t)$ is finite and goes to zero as $n \to \infty$ for any nonnegative integer $k$ and decreases along $k$. Here $f(t)$ also depends on the dimension $d$ and the sample size $n$.

Condition 3.8 states that, in addition to the upper bound on the estimation error defined in Definition 3.5, the sample versions of the score function defined in (3.3) and the deviation function defined in (3.4) as well as their derivatives all well concentrate to their population versions. In particular, the condition on the integrability of the tail function $f(t)$ characterizes that the tail probability of concentration decays faster than the polynomial rate. As we will illustrate using the multinomial logistic regression model in §D, such a superpolynomial tail behavior is typically guaranteed by exponential concentration inequalities such as Bernstein-type inequalities and their matrix variants.

The following theorem compares the mean squared errors of the MLIPS and IPS estimators defined in (3.1) and (2.2). Recall that $\zeta_{DG}$, $\zeta_{ST}$, and $\zeta_{DH}$ are defined in Assumption 3.4. Since $\zeta_{DG}$, $\zeta_{ST}$, and $\zeta_{DH}$ are population quantities, they do not scale with the sample size $n$.

**Theorem 3.9** (Mean Squared Error Reduction). Under Condition 3.8 and Assumptions 3.3 and 3.4, the MLIPS $\widehat{V}$ is asymptotically unbiased and it holds that
$$\mathbb{E}\big[(\widehat{V} - V)^2\big] = \mathbb{E}\big[(\widetilde{V} - V)^2\big] - \Big(\underbrace{1/n \cdot \mathrm{Var}\big(\Pi(r, x, a; \beta^*)\big)}_{\text{Reduction of MSE}} - \xi(n)\Big), \tag{3.13}$$
where
$$\Pi(a, x; \beta^*) = \mathbb{E}\big[D_V(r, a, x; \beta^*) \cdot S(a, x; \beta^*)^\top\big] \cdot I^{-1}(\beta^*) \cdot S(a, x; \beta^*), \tag{3.14}$$
and $\xi(n) = (\mathbb{E}[(\widetilde{V} - V)^2])^{1/2} \cdot O((\zeta_{DG} \cdot (\zeta_{ST} + 1) + \zeta_{DH}) \cdot (\int_0^\infty t^4 \, df(t))^{1/2}) + O((\zeta_{DG}^2 \cdot (\zeta_{ST} + 1) + \zeta_{DG} \cdot \zeta_{DH}) \cdot \int_0^\infty t^3 \, df(t))$.

*Proof.* See §A for a detailed proof. $\square$

It is worth noting that the key to our guarantee for MSE reduction lies in the fact that
$$\mathbb{E}\left[\left((\widetilde{V} - V) - \frac{1}{n} \sum_{i=1}^n \Pi(r_i, a_i, x_i; \beta^*)\right)^2\right] = \mathbb{E}\big[(\widetilde{V} - V)^2\big] - 1/n \cdot \mathrm{Var}\big(\Pi(r, x, a; \beta^*)\big),$$
which is actually an orthogonality relation between $(\widetilde{V} - V) - (1/n) \cdot \sum_{i=1}^n \Pi(r_i, a_i, x_i; \beta^*)$ and $(1/n) \cdot \sum_{i=1}^n \Pi(r_i, a_i, x_i; \beta^*)$.

We interpret the MSE reduction guarantee in Theorem 3.9 as follows. For the simplicity of discussion, for now we assume that the dimension $d$ does not scale with the sample size $n$. The $1/n \cdot \mathrm{Var}(\Pi(r, x, a; \beta^*))$ term on the right-hand side of (3.13) characterizes the reduction of the mean squared error. For example, in the multinomial logistic regression model, the $\xi(n)$ term is roughly of the order $1/n^{3/2}$. As a consequence, for a sufficiently large $n$, we have
$$1/n \cdot \mathrm{Var}\big(\Pi(r, x, a; \beta^*)\big) - \xi(n) \geq 1/(2n) \cdot \mathrm{Var}\big(\Pi(r, x, a; \beta^*)\big). \tag{3.15}$$
In other words, the mean squared error of the MLIPS estimator is at least smaller than that of the IPS estimator by a margin of $1/(2n) \cdot \mathrm{Var}(\Pi(r, x, a; \beta^*))$, where $\mathrm{Var}(\Pi(r, x, a; \beta^*))$ is a population quantity that does not scale with $n$. In comparison, the mean squared error is also of the order $1/n$ in the multinomial logistic regression model. That is to say, the reduction effect does not vanish even when $n \to \infty$. See §D for more discussions and a more detailed illustration of Theorem 3.9 using the multinomial logistic regression model.

## 3.2 PRACTICAL CONSIDERATIONS

Throughout the theoretical analysis, we assume that the parametrization of the logging policy is known. However, in practice we may not know the class of logging policies that generate the logs, in which cases we can use universal function approximators such as neural networks to estimate such surrogate policies. The parametrization of the logging distribution may be misspecified when we

use those approximators. However, when universal function approximators such as neural networks are used, the approximation error often diminishes with an increasing number of layers and neurons (see, for example, Schmidt-Hieber (2017); Telgarsky (2016); Yarotsky (2017)). Such a diminishing approximation error should enter the Taylor expansions in Lemma A.1-A.2.

In our experiments, we found that neural network policies worked very well on both standard multi-label classification datasets and a large-scale ad placement dataset.

Due to its simplicity, MLIPS can be easily implemented without much extra effort. Furthermore, MLIPS is orthogonal to other existing approaches for mean squared error reduction in policy evaluation and optimization. Hence, it can often be combined straightforwardly with them. In our experiments, we observed that it can be used to further improve POEM for policy optimization.

Another advantage of our approach is that it still works even when we have no access to the logging policy, which often happens in real world due to various reasons. In those cases, IPS may not work at all, but our approach exhibits superior performance, as shown in our experiments.

## 4 EXPERIMENTS

In this section, we empirically evaluate the effectiveness of the proposed method on a number of datasets. We first check the reduction of the mean squared error enabled by the MLIPS estimator. Then we apply IPS, MLIPS, POEM and MLPOEM to policy optimization and compare their performance on several datasets.

### 4.1 SETTINGS

**Datasets.** First, we adopt the same experimental design as the one specified in Swaminathan & Joachims (2015a) to generate batch contextual bandit learning datasets from supervised learning datasets. Four multi-label datasets are selected from the LibSVM repository (see Swaminathan & Joachims (2015a) for more details). To construct a batch contextual bandit learning dataset, we take a supervised learning dataset $D^* = \{(x_i, y_i^*)\}_{i=1}^n$, simulate using a logging policy $\mu$ by sampling $y_i \sim \mu(\cdot \,|\, x_i)$, and collect the reward as the negative Hamming distance $-\Delta(y_i^*, y_i)$ between the supervised label and the sampled label. We use a Conditional Random Field (CRF) model (Sutton & McCallum, 2012) trained on 5% of the data as the logging policy $\mu$. We repeat this procedure four times to generate the entire bandit feedback dataset $D = \{(x_i, y_i, r_i = -\Delta(y_i^*, y_i), \mu(y_i \,|\, x_i))\}_{i=1}^n$. Both the training sets and held-out testing sets are constructed in the same way as suggested in Swaminathan & Joachims (2015a).

In addition to these standard datasets, we also evaluate the effectiveness of our method on a large-scale ad placement dataset. The dataset is collected by Criteo, a leader in the display advertising industry (Lefortier et al., 2016). For this dataset, we consider an ad display scenario where a policy selects the products to be displayed on a website when a user arrives. Provided the candidate set of products and impression context, we hope the policy to place ads such that the aggregated click-through-rate (CTR) or the number of clicks is maximized. We consider a subset of the dataset, which is used in the NIPS 2017 Criteo Ad Placement Challenge. Each ad is represented by a $74000$-dimensional feature vector. For each context (or impression), the logged action of selecting a candidate ad, a binary reward value on whether the displayed product is clicked by the user, and the probability that the logging policy picks such a specific candidate are recorded. More details can be found at CrowdAI (2017).

**Surrogate policy estimation.** To fit the surrogate policy, we apply two models, a simple logistic regression model and a one-hidden-layer perceptron with ten ReLU activation units, both with $\ell_2$-regularization. Here five-fold cross validation is used to tune the regularization parameter. We use the L-BFGS algorithm (Liu & Nocedal, 1989) for optimization.

**Policy optimization.** We consider IPS, POEM (Swaminathan & Joachims, 2015a), and Norm-POEM (Swaminathan & Joachims, 2015b). We compare their performance with MLIPS as well as the combination of our surrogate policy technique with POEM (MLPOEM). There are several hyperparameters in POEM, Norm-POEM, and MLPOEM, including the thresholding parameter and the regularization parameter. We use five-fold cross validation on the training set to tune them. For reward maximization, following the procedures of POEM, we initialize the model weights to zeros,

use mini-batch AdaGrad (Duchi et al., 2011) with a batch size of $100$ and step size $\eta = 1$. On all datasets, we repeat the experiments ten times to report the average generalization performance.

## 4.2 Mean Squared Error Reduction in Policy Estimation

First, we check whether the MLIPS estimator has a smaller mean squared error than the vanilla IPS estimator. Instead of comparing the estimation of the value function, we focus on comparing the performance of MLIPS and IPS on off-policy gradient estimation, whose accuracy is crucial to policy optimization. We use the LYRL dataset (Swaminathan & Joachims, 2015a) and combine the training and testing sets to obtain a large bandit feedback dataset. Given a reasonably accurate policy weight vector (after training with $10000$ mini-batch updates), we compute its IPS-based gradient estimator on the entire dataset to get a rough "ground-truth" gradient. We further construct a series of data subsets to evaluate the gradient estimation with smaller sample sizes, with $2^{-1/2}, 2^{-2/2}, \ldots, 2^{-k/2}$ of the entire dataset.

For the neural network estimate of surrogate policy (NN-surrogate), we first fit a logistic regression model and a one-hidden-layer perceptron with ten ReLU activation units to estimate the surrogate policy. After this step, we compute the Euclidean distance between the "ground-truth" gradient and the gradient estimators that correspond to smaller sample sizes. Average Euclidean distances and standard deviations are computed over twenty trials. Results are illustrated in Figure 1.

We observe that the MLIPS estimator gives much better gradient estimation than the original IPS estimator. In particular, the relative improvement is significant especially when the sample size is small. It is also interesting to see that the difference between the linear surrogate policy and the neural network surrogate policy is very small, possibly because the actual logging CRF policy is linear. On the Criteo dataset, in which the logging policy is unknown and possibly very complicated, we find that the neural network surrogate policy works better than the linear policy.

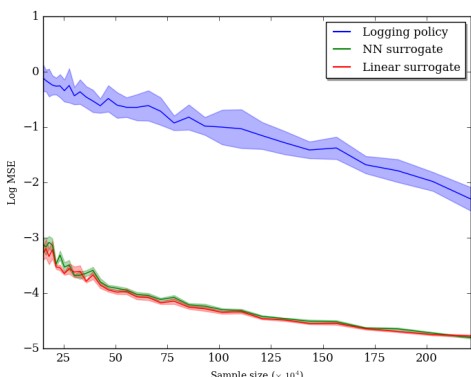

Figure 1: Log MSE of estimations for off-policy gradients made by IPS (logging policy), MLIPS (linear surrogate) and NN-fitted MLIPS (NN-surrogate) for logistic regression model on the LYRL dataset in Swaminathan & Joachims (2015a).

## 4.3 Generalization Performance of Policy Optimization

To evaluate the effectiveness of our method on policy optimization, we train logistic regression policies with IPS, POEM, Norm-POEM objective functions, and the IPS and POEM versions with maximum likelihood surrogate policies (MLIPS and MLPOEM). As mentioned before, we use both the logistic regression model and the one-hidden-layer perceptron model to fit the surrogate policies, which are respectively marked with "-Lin" and "-NN" suffixes in the names of the training methods. Results are summarized in Table 1. The test set Hamming losses for all methods are averaged over ten runs. On each run, both MLIPS and MLPOEM consistently and significantly outperforms their corresponding original versions.

As a control, we study the scenario where the logging policy is not available. We train the policies using IPS while treating the logging policy to be uniform. The performance of such a variant of IPS (the last row in the table with algorithm name "IPS-Uniform") is significantly worse than our method, which also does not require knowing the logging policy. Notably, when compared against (Norm-)POEM, the state-of-the-art algorithm on these datasets, our MLPOEM algorithm can augment the generalization performance on most datasets. This indicates that our surrogate policy technique provides a complementary mean squared error reduction effect for existing methods in this field. Finally, it is worth noting that the methods with neural network surrogate polices and linear policies are not too much different in terms of performance on these datasets, possibly because the logging policies used to generate these datasets are linear.

### 4.3.1 CRITEO AD PLACEMENT DATASET

We also evaluate the performance of our method on the preprocessed data used in the NIPS 2017 Workshop Criteo ad placement challenge. Because there is no information about the logging policy used to generate the data, it is reasonable to use a complex universal function approximator to fit the surrogate policy. We implement a neural network model with 21 tanh activation units for both the surrogate policy and the target policy. Due to the massive size of this dataset and its high-dimensional and sparse feature representation, we implement the $\ell_2$-regularized IPS and MLIPS in C++ with multi-threading.

Both the maximum likelihood estimation and policy optimization are performed using the L-BFGS algorithm. We split the dataset into training (80%) and testing (20%) sets. The regularization parameter is chosen by five-fold cross-validation on the training set. Evaluation is performed by computing the inverse propensity weights as suggested by the NIPS challenge. In comparison with IPS, which obtains $0.556$ on the training set and $0.551$ on the testing set, our method obtains $0.586$ on the training set and $0.572$ on the testing set. An implementation of our method ($0.576$ on the training set and $0.568$ on the testing set, which are evaluated on the hold out data for the competition) is also ranked among very top (prize winning teams) of the NIPS 2017 challenge leaderboard.

Table 1: Evaluations of the Hamming losses for policy optimizations performed by importance weighed policy gradient estimators and their ML-variants on the datasets of Swaminathan & Joachims (2015a). Evaluations of Norm-POEM are also included as general benchmarks. We include extra results for this experiment in Appendix E

| Dataset/Alg. | Scene | Yeast | TMC | LYRL |
|---|---|---|---|---|
| IPS | 1.342 | 4.571 | 3.023 | 1.108 |
| POEM | 1.143 | 4.549 | 2.522 | 0.981 |
| Norm-POEM | 1.045 | 3.876 | 2.072 | 0.799 |
| MLIPS-Lin | 1.086 | 3.778 | 2.018 | 1.025 |
| MLIPS-NN | 1.086 | 3.630 | 2.019 | 0.930 |
| MLPOEM-Lin | 1.086 | 3.894 | 2.010 | 0.949 |
| MLPOEM-NN | 1.086 | 3.477 | 2.000 | 0.904 |
| IPS-Uniform | 1.086 | 5.893 | 6.174 | 1.463 |

## 5 CONCLUSION

We introduce a new but simple approach for mean squared error reduction in policy evaluation and optimization. Theoretical analysis illustrates that the proposed MLIPS estimator is asymptotically unbiased, and moreover, has a smaller mean squared error than the classical IPS estimator. Experimental results on several large-scale datasets also demonstrate the empirical effectiveness of the proposed estimator. Our technique can also be combined with existing approaches and further improves the performance.

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

## A    PROOF OF THEOREM 3.9

The proof of Theorem 3.9 is established based on the following two lemmas. First, we introduce a lemma that relates the errors of the MLIPS and IPS estimators. Recall that, as defined in Definitions 3.5-3.7 and Condition 3.8, we have $t^\star = (t_\beta, t_S, t_D)$ where $t_S = (t_{SG}, t_{SH}, t_{ST})$ and $t_D = (t_{DG}, t_{DH})$. Also, recall that $\zeta_{DG}, \zeta_{DH}$, and $\zeta_{ST}$ are defined in Assumption 3.4.

**Lemma A.1.** For the MLIPS estimator $\widehat{V}$ defined in (3.1) and the IPS estimator $\widetilde{V}$ defined in (2.2), under Assumption 3.4 and conditioning on the event $\mathcal{E}(t^\star)$ defined in (3.12), we have

$$\widehat{V} - V = (\widetilde{V} - V) - \mathbb{E}\big[D_V(r, a, x; \beta^*) \cdot S(a, x; \beta^*)^\top\big] \cdot (\widehat{\beta} - \beta^*)$$
$$+ O\big(t_\beta^2 \cdot (t_{DH} + \zeta_{DH}) + t_\beta \cdot t_{DG}\big). \tag{A.1}$$

*Proof.* See §B for a detailed proof.  $\square$

The following lemma characterizes the $\widehat{\beta} - \beta^*$ term on the right-hand side of (A.1), which is the estimation error of $\widehat{\beta}$.

**Lemma A.2.** For the maximum likelihood estimator $\widehat{\beta}$ and the true parameter $\beta^*$, we have
$$\widehat{\beta} - \beta^*$$
$$= I^{-1}(\beta^*) \cdot \left[\frac{1}{n}\sum_{i=1}^{n} S(a_i, x_i; \beta^*) - \Delta_S(\widehat{\beta} - \beta^*) + \frac{1}{2n}\sum_{i=1}^{n} \frac{\partial^2 S(a_i, x_i; \beta_S)}{\partial \beta^2}(\widehat{\beta} - \beta^*, \widehat{\beta} - \beta^*, \cdot)\right],$$

where

$$\Delta_S = \frac{1}{n}\sum_{i=1}^{n} \frac{\partial S(a_i, x_i; \beta^*)}{\partial \beta} - \mathbb{E}\left[\frac{\partial S(a, x; \beta^*)}{\partial \beta}\right], \quad \beta_S = \lambda_S \beta^* + (1 - \lambda_S)\widehat{\beta}. \tag{A.2}$$

Here $\lambda_S \in [0, 1]$, $I^{-1}(\beta^*)$ is the inverse of the Fisher information matrix defined in (3.3), and $(\partial^2 S(a, x; \beta^*)/\partial \beta^2)(v_1, v_2, \cdot)$ is the bilinear map defined in Assumption 3.4.

*Proof.* See §C for a detailed proof.  $\square$

Lemma A.2 allows us to more precisely characterize the $\widehat{\beta} - \beta^*$ term in (A.1). Now we are ready to prove Theorem 3.9.

*Proof of Theorem 3.9.* By combining Lemmas A.1 and A.2, under Assumptions 3.3 and 3.4 and conditioning on the event $\mathcal{E}(t^\star)$ defined in (3.12), we have

$$\widehat{V} - V = (\widetilde{V} - V) - \frac{1}{n}\sum_{i=1}^{n}\underbrace{\mathbb{E}\big[D_V(r, a, x; \beta^*) \cdot S(a, x; \beta^*)^\top\big] \cdot I^{-1}(\beta^*) \cdot S(a_i, x_i; \beta^*)}_{\Pi(r_i, a_i, x_i; \beta^*)} + \eta(t^\star),$$
$$\tag{A.3}$$

where

$$\eta(t^\star) = O\big(\zeta_{DG} \cdot t_\beta \cdot (\zeta_{ST} \cdot t_\beta + t_{SH}) + t_\beta^2 \cdot (t_\beta \cdot t_{DG} + t_{DH} + \zeta_{DH})\big). \tag{A.4}$$

First, we show that the MLIPS $\widetilde{V}$ is asymptotically unbiased. Note that we have $\mathbb{E}[S(a, x; \beta^*)] = 0$, the expectation of the second term on the right hand side of (A.3) is zero. Thus, as $\widetilde{V}$ is an unbiased estimator of $V$, we have

$$\lim_{n \to \infty} \mathbb{E}[\widehat{V} - V] = \lim_{n \to \infty} \mathbb{E}[\eta(t^\star)]. \tag{A.5}$$

By Condition 3.8 and the definition of $\eta(t^\star)$ in (A.4), we have

$$\mathbb{E}\big[\eta(t^\star) \,\big|\, \mathcal{E}(t^\star)\big] = O\big(\zeta_{DG} \cdot t_\beta \cdot (\zeta_{ST} \cdot t_\beta + t_{SH}) + t_\beta^2 \cdot (t_{DH} + \zeta_{DH} + t_\beta \cdot t_{DG})\big)$$
$$= O\big(\zeta_{DG} \cdot (\zeta_{ST} + 1) \cdot t^2 + t^2 \cdot (t^2 + t + \zeta_{DH})\big), \tag{A.6}$$

where $t = \|t^\star\|_\infty$. As defined in Condition 3.8, $\int_0^\infty t^k \, \mathrm{d}f(t)$ goes to zero when $n \to \infty$ for any $k$. Therefore, when $n \to \infty$, from (A.6) we have

$$\mathbb{E}[\eta(t^\star)] = \mathbb{E}\Big[\mathbb{E}\big[\eta(t^\star) \,\big|\, \mathcal{E}(t^\star)\big]\Big] = O\bigg((\zeta_{\mathrm{DG}} \cdot (\zeta_{\mathrm{ST}} + 1) + \zeta_{\mathrm{DH}}) \cdot \int_0^\infty t^2 \, \mathrm{d}f(t)\bigg), \qquad \text{(A.7)}$$

which gives $\lim_{n \to \infty} \mathbb{E}[\eta(t^\star)] = 0$. Thus, we have $\lim_{n \to \infty}[\widehat{V} - V] = 0$, which means that the MLIPS estimator $\widehat{V}$ is asymptotically unbiased.

Next, we proceed to prove the MSE reducing effect of $\widehat{V}$. For the mean squared error of the MLIPS estimator $\widehat{V}$, we have

$$\mathbb{E}\big[(\widehat{V} - V)^2\big] = \mathbb{E}\bigg[\bigg((\widetilde{V} - V) - \frac{1}{n}\sum_{i=1}^n \Pi(r_i, a_i, x_i; \beta^*) + \eta(t^\star)\bigg)^2\bigg]$$

$$= \underbrace{\mathbb{E}\bigg[\bigg((\widetilde{V} - V) - \frac{1}{n}\sum_{i=1}^n \Pi(r_i, a_i, x_i; \beta^*)\bigg)^2\bigg]}_{\text{(i)}} + \underbrace{\mathbb{E}\big[\mathbb{E}\big[\eta^2(t^\star) \,\big|\, \mathcal{E}(t^\star)\big]\big]}_{\text{(ii)}}$$

$$+ 2 \cdot \underbrace{\mathbb{E}\bigg[\bigg((\widetilde{V} - V) - \frac{1}{n}\sum_{i=1}^n \Pi(r_i, a_i, x_i; \beta^*)\bigg) \cdot \eta(t^\star)\bigg]}_{\text{(iii)}}, \qquad \text{(A.8)}$$

where for term (ii) we use the law of total expectation. In the sequel, we characterize terms (i)-(iii) respectively.

**Term (i) in** (A.8)**:** Note that by Definition 3.2 we have

$$\widetilde{V} - V = \frac{1}{n}\sum_{i=1}^n D_V(r_i, a_i, x_i; \beta^*), \quad \text{where} \quad D_V(r, a, x; \beta) = \frac{\pi(a, x)}{\mu(a, x; \beta)} \cdot r(a, x) - V.$$

We reformulate the first term on the right-hand side of (A.8) as

$$\mathbb{E}\bigg[\bigg((\widetilde{V} - V) - \frac{1}{n}\sum_{i=1}^n \Pi(r_i, a_i, x_i; \beta^*)\bigg)^2\bigg] = \mathbb{E}\big[(\widetilde{V} - V)^2\big] + \mathbb{E}\bigg[\bigg(\frac{1}{n}\sum_{i=1}^n \Pi(r_i, a_i, x_i; \beta^*)\bigg)^2\bigg]$$

$$- 2/n^2 \cdot \mathbb{E}\bigg[\bigg(\sum_{i=1}^n D_V(r_i, a_i, x_i; \beta^*)\bigg) \cdot \bigg(\sum_{i=1}^n \Pi(r_i, a_i, x_i; \beta^*)\bigg)\bigg]. \qquad \text{(A.9)}$$

Then by the fact that $\mathbb{E}[S(a, x; \beta^*)] = 0$, for the second term on the right-hand side of (A.9) we have

$$\mathbb{E}\big[\Pi(r, a, x; \beta^*)\big] = \mathbb{E}\big[D_V(r, a, x; \beta^*) \cdot S(a, x; \beta^*)^\top\big] \cdot I^{-1}(\beta^*) \cdot \mathbb{E}\big[S(a, x; \beta^*)\big] = 0. \quad \text{(A.10)}$$

Meanwhile, we have

$$\mathbb{E}\bigg[\bigg(\sum_{i=1}^n D_V(r_i, a_i, x_i; \beta^*)\bigg) \cdot \bigg(\sum_{i=1}^n \Pi(r_i, a_i, x_i; \beta^*)\bigg)\bigg]$$

$$= \sum_{i=1}^n \mathbb{E}\big[D_V(r_i, a_i, x_i; \beta^*) \cdot \Pi(r_i, a_i, x_i; \beta^*)\big] + \sum_{i \neq j} \mathbb{E}\big[D_V(r_i, a_i, x_i; \beta^*) \cdot \Pi(r_j, a_j, x_j; \beta^*)\big]$$

$$= n \cdot \mathbb{E}\big[D_V(r, a, x; \beta^*) \cdot \Pi(r, a, x; \beta^*)\big], \qquad \text{(A.11)}$$

where the second equality follows from (A.10). Hence, we obtain from (A.9) and (A.11) that

$$\mathbb{E}\bigg[\bigg((\widetilde{V} - V) - \frac{1}{n}\sum_{i=1}^n \Pi(r_i, a_i, x_i; \beta^*)\bigg)^2\bigg] \qquad \text{(A.12)}$$

$$= \mathbb{E}\big[(\widetilde{V} - V)^2\big] + 1/n \cdot \mathrm{Var}\big(\Pi(r, x, a; \beta^*)\big) - 2/n^2 \cdot n \cdot \mathbb{E}\big[D_V(r, a, x; \beta^*) \cdot \Pi(r, a, x; \beta^*)\big].$$

Note that we have the following equality for the Fisher information matrix $I(\beta^*)$,

$$\mathbb{E}\big[S(a, x; \beta^*) \cdot S(a, x; \beta^*)^\top\big] = I(\beta^*) = -\partial S(a, x; \beta^*)/\partial \beta. \qquad \text{(A.13)}$$

Then by the definition of $\Pi(r, a, x; \beta^*)$ in (3.14), we have

$$\mathbb{E}\big[\Pi^2(r, a, x; \beta^*)\big] = \mathbb{E}\big[D_V(r, a, x; \beta^*) \cdot S(a, x; \beta^*)^\top\big] \cdot I^{-1}(\beta^*) \cdot \mathbb{E}\big[S(a, x; \beta^*) \cdot S(a, x; \beta^*)^\top\big]$$
$$\cdot I^{-1}(\beta^*) \cdot \mathbb{E}\big[D_V(r, a, x; \beta^*) \cdot S(a, x; \beta^*)\big],$$

which by (A.13) and the definition of $\Pi(r, a, x; \beta^*)$ implies

$$\mathbb{E}\big[\Pi^2(r, a, x; \beta^*)\big]$$
$$= \mathbb{E}\big[D_V(r, a, x; \beta^*) \cdot S(a, x; \beta^*)^\top\big] \cdot I^{-1}(\beta^*) \cdot \mathbb{E}\big[D_V(r, a, x; \beta^*) \cdot S(a, x; \beta^*)\big]$$
$$= \mathbb{E}\big[D_V(r, a, x; \beta^*) \cdot \Pi(r, a, x; \beta^*)\big]. \tag{A.14}$$

Note that (A.14) implies for the third term on the right-hand side of (A.12), we have

$$\mathbb{E}\big[D_V(r, a, x; \beta^*) \cdot \Pi(r, a, x; \beta^*)\big] = \mathbb{E}\big[\Pi^2(r, a, x; \beta^*)\big] = \mathrm{Var}\big(\Pi(r, x, a; \beta^*)\big), \tag{A.15}$$

where the second equality follows from (A.10). Plugging (A.15) into (A.12), we obtain

$$\mathbb{E}\Bigg[\bigg((\widetilde{V} - V) - \frac{1}{n}\sum_{i=1}^{n}\Pi(r_i, a_i, x_i; \beta^*)\bigg)^2\Bigg] = \mathbb{E}\big[(\widetilde{V} - V)^2\big] - 1/n \cdot \mathrm{Var}\big(\Pi(r, x, a; \beta^*)\big). \tag{A.16}$$

**Term (ii) in** (A.8)**:** By Condition 3.8 and the definition of $\eta(t^\star)$ in (A.4), we have

$$\mathbb{E}\big[\eta^2(t^\star) \,\big|\, \mathcal{E}(t^\star)\big] = O\Big(\big(\zeta_{\mathrm{DG}} \cdot t_\beta \cdot (\zeta_{\mathrm{ST}} \cdot t_\beta + t_{\mathrm{SH}}) + t_\beta^2 \cdot (t_{\mathrm{DH}} + \zeta_{\mathrm{DH}} + t_\beta \cdot t_{\mathrm{DG}})\big)^2\Big)$$
$$= O\Big(\big(\zeta_{\mathrm{DG}} \cdot (\zeta_{\mathrm{ST}} + 1) \cdot t^2 + t^2 \cdot (t^2 + t + \zeta_{\mathrm{DH}})\big)^2\Big), \tag{A.17}$$

where $t = \|t^\star\|_\infty$. As defined in Condition 3.8, $\int_0^\infty t^k \, \mathrm{d}f(t)$ is a decreasing function of $k$. Therefore, from (A.17) we have

$$\mathbb{E}\Big[\mathbb{E}\big[\eta^2(t^\star) \,\big|\, \mathcal{E}(t^\star)\big]\Big] = O\bigg(\big(\zeta_{\mathrm{DG}} \cdot (\zeta_{\mathrm{ST}} + 1) + \zeta_{\mathrm{DH}}\big)^2 \cdot \int_0^\infty t^4 \, \mathrm{d}f(t)\bigg). \tag{A.18}$$

**Term (iii) in** (A.8)**:** By the law of total expectation, we have

$$\mathbb{E}\Bigg[\bigg((\widetilde{V} - V) - \frac{1}{n}\sum_{i=1}^{n}\Pi(r_i, a_i, x_i; \beta^*)\bigg) \cdot \eta(t^\star)\Bigg]$$
$$= \mathbb{E}\big[(\widetilde{V} - V) \cdot \eta(t^\star)\big] - \mathbb{E}\Bigg[\mathbb{E}\bigg[\bigg(\frac{1}{n}\sum_{i=1}^{n}\Pi(r_i, a_i, x_i; \beta^*)\bigg) \cdot \eta(t^\star) \,\bigg|\, \mathcal{E}(t^\star)\bigg]\Bigg]. \tag{A.19}$$

For the first term on the right-hand side of (A.19), by the Cauchy-Schwartz inequality for expectation, we have

$$\mathbb{E}\big[(\widetilde{V} - V) \cdot \eta(t^\star)\big] \le \bigg(\mathbb{E}\big[(\widetilde{V} - V)^2\big] \cdot \mathbb{E}\Big[\mathbb{E}\big[\eta^2(t^\star) \,\big|\, \mathcal{E}(t^\star)\big]\Big]\bigg)^{1/2} \tag{A.20}$$
$$= \big(\mathbb{E}\big[(\widetilde{V} - V)^2\big]\big)^{1/2} \cdot O\bigg(\big(\zeta_{\mathrm{DG}} \cdot (\zeta_{\mathrm{ST}} + 1) + \zeta_{\mathrm{DH}}\big) \cdot \bigg(\int_0^\infty t^4 \, \mathrm{d}f(t)\bigg)^{1/2}\bigg),$$

where the second equality follows from (A.18). Now we consider the second term on the right-hand side of (A.19). Note that by the definition of $\Pi(r, a, x; \beta)$ in (A.3), we have

$$\frac{1}{n}\sum_{i=1}^{n}\Pi(r_i, a_i, x_i; \beta^*) = \frac{1}{n}\sum_{i=1}^{n}\mathbb{E}\big[D_V(r, a, x; \beta^*) \cdot S(a, x; \beta^*)^\top\big] \cdot I^{-1}(\beta^*) \cdot S(a_i, x_i; \beta^*)$$
$$\le \big\|\mathbb{E}\big[D_V(r, a, x; \beta^*) \cdot S(a, x; \beta^*)^\top\big]\big\| \cdot \|I^{-1}(\beta^*)\| \cdot \bigg\|\frac{1}{n}\sum_{i=1}^{n}S(a_i, x_i; \beta^*)\bigg\|, \tag{A.21}$$

where by (A.23) we have that, conditioning on the event $\mathcal{E}(t^\star)$ defined in (3.12),

$$\bigg\|\frac{1}{n}\sum_{i=1}^{n}S(a_i, x_i; \beta^*)\bigg\| \le t_{\mathrm{SG}}.$$

Under Assumption 3.4, we prove in Lemma B.1 and (B.2) that

$$\left\| \mathbb{E}\big[D_V(r,a,x;\beta^*) \cdot S(a,x;\beta^*)^\top\big]\right\| = \left\|\mathbb{E}\left[\frac{\partial D_V(r,a,x;\beta^*)}{\partial \beta}\right]\right\| = O(\zeta_{\mathrm{DG}}). \tag{A.22}$$

Note that in the event $\mathcal{E}_{\mathrm{SG}}(t_{\mathrm{SG}})$ defined in (3.9), we have $\mathbb{E}[S(a,x;\beta^*)] = 0$. Hence, equivalently we have

$$\mathcal{E}_{\mathrm{SG}}(t_{\mathrm{SG}}) = \left\{\left\|\frac{1}{n}\sum_{i=1}^n S(a_i,x_i;\beta^*)\right\| \le t_{\mathrm{SG}}\right\}. \tag{A.23}$$

Combining (A.4), (A.21), (A.22), (A.23), and Assumption 3.3, under Condition 3.8 we have

$$\mathbb{E}\left[\left(\frac{1}{n}\sum_{i=1}^n \Pi(r_i,a_i,x_i;\beta^*)\right) \cdot \eta(t^\star)\,\Big|\,\mathcal{E}(t^\star)\right]$$

$$= O\Big(\zeta_{\mathrm{DG}} \cdot t_{\mathrm{SG}} \cdot \big(\zeta_{\mathrm{DG}} \cdot t_\beta \cdot (\zeta_{\mathrm{ST}} \cdot t_\beta + t_{\mathrm{SH}}) + t_\beta^2 \cdot (t_\beta \cdot t_{\mathrm{DG}} + t_{\mathrm{DH}} + \zeta_{\mathrm{DH}})\big)\Big)$$

$$= O\Big(\zeta_{\mathrm{DG}}^2 \cdot (\zeta_{\mathrm{ST}} + 1) \cdot t^3 + \zeta_{\mathrm{DG}} \cdot t^3 \cdot (t^2 + t + \zeta_{\mathrm{DH}})\Big), \tag{A.24}$$

where $t = \|t^\star\|_\infty$. By (A.24), using a similar argument to the one used to obtain (A.18), we have

$$\mathbb{E}\left[\mathbb{E}\left[\left(\frac{1}{n}\sum_{i=1}^n \Pi(r_i,a_i,x_i;\beta^*)\right) \cdot \eta(t^\star)\,\Big|\,\mathcal{E}(t^\star)\right]\right]$$

$$= O\left(\big(\zeta_{\mathrm{DG}}^2 \cdot (\zeta_{\mathrm{ST}} + 1) + \zeta_{\mathrm{DG}} \cdot \zeta_{\mathrm{DH}}\big) \cdot \int_0^\infty t^3\,\mathrm{d}f(t)\right). \tag{A.25}$$

Thus, combining (A.20) and (A.25), for (A.19) we obtain

$$\mathbb{E}\left[\left((\widetilde{V} - V) - \frac{1}{n}\sum_{i=1}^n \Pi(r_i,a_i,x_i;\beta^*)\right) \cdot \eta(t^\star)\right]$$

$$= \big(\mathbb{E}[(\widetilde{V} - V)^2]\big)^{1/2} \cdot O\left(\big(\zeta_{\mathrm{DG}} \cdot (\zeta_{\mathrm{ST}} + 1) + \zeta_{\mathrm{DH}}\big) \cdot \left(\int_0^\infty t^4\,\mathrm{d}f(t)\right)^{1/2}\right)$$

$$+ O\left(\big(\zeta_{\mathrm{DG}}^2 \cdot (\zeta_{\mathrm{ST}} + 1) + \zeta_{\mathrm{DG}} \cdot \zeta_{\mathrm{DH}}\big) \cdot \int_0^\infty t^3\,\mathrm{d}f(t)\right). \tag{A.26}$$

Note that term (ii) in (A.8), which is characterized by (A.18), is dominated by the last term on the right-hand side of (A.25), since $\int_0^\infty t^k\,\mathrm{d}f(t)$ decreases along $k$ under Condition 3.8. Plugging (A.16), (A.18), and (A.26) into (A.8), we obtain

$$\mathbb{E}[(\widehat{V} - V)^2] = \mathbb{E}[(\widetilde{V} - V)^2] - 1/n \cdot \mathrm{Var}\big(\Pi(r,x,a;\beta^*)\big)$$

$$+ \big(\mathbb{E}[(\widetilde{V} - V)^2]\big)^{1/2} \cdot O\left(\big(\zeta_{\mathrm{DG}} \cdot (\zeta_{\mathrm{ST}} + 1) + \zeta_{\mathrm{DH}}\big) \cdot \left(\int_0^\infty t^4\,\mathrm{d}f(t)\right)^{1/2}\right)$$

$$+ O\left(\big(\zeta_{\mathrm{DG}}^2 \cdot (\zeta_{\mathrm{ST}} + 1) + \zeta_{\mathrm{DG}} \cdot \zeta_{\mathrm{DH}}\big) \cdot \int_0^\infty t^3\,\mathrm{d}f(t)\right),$$

which concludes the proof of Theorem 3.9. $\qquad\square$

## B  PROOF OF LEMMA A.1

Before proving the Lemma A.1, we first introduce the following lemma on the gradient of the deviation function $D_V(r,a,x;\beta)$.

**Lemma B.1.** For the score and deviation functions defined in (3.3) and (3.4), we have

$$\mathbb{E}\left[\frac{\partial D_V(r,a,x;\beta^*)}{\partial \beta}\right] = -\mathbb{E}\big[D_V(r,a,x;\beta^*) \cdot S(a,x;\beta^*)\big].$$

*Proof.* By Definition 3.2, we have

$$\mathbb{E}\left[\frac{\partial D_V(r,a,x;\beta^*)}{\partial \beta}\right] = -\mathbb{E}\left[\frac{\pi(a,x) \cdot r(a,x)}{\mu^2(a,x;\beta^*)} \cdot \frac{\partial \mu(a,x;\beta^*)}{\partial \beta}\right]. \tag{B.1}$$

Then using the fact that $\mathbb{E}\big[S(a, x; \beta^*)\big] = 0$, we have

$$-\mathbb{E}\big[D_V(r, a, x; \beta^*) \cdot S(a, x; \beta^*)\big] = \mathbb{E}\left[\left(V - \frac{\pi(a, x)}{\mu(a, x; \beta^*)} \cdot r(a, x)\right) \cdot S(a, x; \beta^*)\right]$$

$$= -\mathbb{E}\left[\frac{\pi(a, x) \cdot r(a, x)}{\mu^2(a, x; \beta^*)} \cdot \frac{\partial \mu(a, x; \beta^*)}{\partial \beta}\right] + V \cdot \mathbb{E}\big[S(a, x; \beta^*)\big] = \mathbb{E}\left[\frac{\partial D_V(r, a, x; \beta^*)}{\partial \beta}\right],$$

where the second equality follows from (3.3) and the last equality follows from (B.1). Hence, we conclude the proof of Lemma B.1. $\qquad\square$

Note that by Lemma B.1 and Assumption 3.4, we also have the following equality,

$$\big\|\mathbb{E}\big[D_V(r, a, x; \beta^*) \cdot S(a, x; \beta^*)\big]\big\| = O(\zeta_{\mathrm{DG}}). \tag{B.2}$$

Now we proceed to prove Lemma A.1.

*Proof of Lemma A.1.* Recall that we have

$$D_V(r, a, x; \beta) = \frac{\pi(a, x)}{\mu(a, x; \beta)} \cdot r(a, x) - V,$$

by which we have

$$\widetilde{V} - V = \frac{1}{n}\sum_{i=1}^{n} D_V(r_i, a_i, x_i; \beta^*), \quad \widehat{V} - V = \frac{1}{n}\sum_{i=1}^{n} D_V(r_i, a_i, x_i; \widehat{\beta}). \tag{B.3}$$

Then by applying the Taylor expansion to the second equality in (B.3), we have

$$\widehat{V} - V = \frac{1}{n}\sum_{i=1}^{n} D_V(r_i, a_i, x_i; \beta^*) + \left[\frac{1}{n}\sum_{i=1}^{n} \frac{\partial D_V(r_i, a_i, x_i; \beta^*)}{\partial \beta}\right]^{\top} \cdot (\widehat{\beta} - \beta^*)$$

$$+ \frac{1}{2n}\sum_{i=1}^{n} (\widehat{\beta} - \beta^*)^{\top} \cdot \frac{\partial^2 D_V(r_i, a_i, x_i; \beta_{\mathrm{D}})}{\partial \beta^2} \cdot (\widehat{\beta} - \beta^*)$$

$$= \widetilde{V} - V + \left[\frac{1}{n}\sum_{i=1}^{n} \frac{\partial D_V(r_i, a_i, x_i; \beta^*)}{\partial \beta}\right]^{\top} \cdot (\widehat{\beta} - \beta^*) + O\big(t_\beta^2 \cdot (t_{\mathrm{DH}} + \zeta_{\mathrm{DH}})\big), \tag{B.4}$$

where $\beta_{\mathrm{D}} = \lambda_{\mathrm{D}}\beta^* + (1 - \lambda_{\mathrm{D}})\widehat{\beta}$ for some $\lambda_{\mathrm{D}} \in [0, 1]$, and the last equality follows from Assumption 3.4 and the fact that we condition on the event $\mathcal{E}(t^\star)$ defined in (3.12). By Lemma B.1, for the second term on the right-hand side of (B.4), we have

$$\left[\frac{1}{n}\sum_{i=1}^{n} \frac{\partial D_V(r_i, a_i, x_i; \beta^*)}{\partial \beta}\right]^{\top} \cdot (\widehat{\beta} - \beta^*)$$

$$= -\mathbb{E}\big[D_V(r, a, x; \beta^*) \cdot S(a, x; \beta^*)^{\top}\big] \cdot (\widehat{\beta} - \beta^*) + \delta_{\mathrm{D}}^{\top}(\widehat{\beta} - \beta^*), \tag{B.5}$$

where

$$\delta_{\mathrm{D}} = \frac{1}{n}\sum_{i=1}^{n} \frac{\partial D_V(r_i, a_i, x_i; \beta^*)}{\partial \beta} - \mathbb{E}\left[\frac{\partial D_V(r, a, x; \beta^*)}{\partial \beta}\right].$$

Meanwhile, conditioning on the event $\mathcal{E}(t^\star)$ defined in (3.12), we have

$$\delta_{\mathrm{D}}^{\top}(\widehat{\beta} - \beta^*) = O(t_\beta \cdot t_{\mathrm{DG}}). \tag{B.6}$$

Combining (B.6), (B.5), and (B.4), we have

$$\widehat{V} - V = (\widetilde{V} - V) - \mathbb{E}\big[D_V(r, a, x; \beta^*) \cdot S(a, x; \beta^*)^{\top}\big] \cdot (\widehat{\beta} - \beta^*)$$

$$+ O\big(t_\beta \cdot t_{\mathrm{DG}} + t_\beta^2 \cdot (t_{\mathrm{DH}} + \zeta_{\mathrm{DH}})\big),$$

which concludes the proof of Lemma A.1. $\qquad\square$

## C    PROOF OF LEMMA A.2

*Proof.* First, recall that we have

$$S(a, x; \beta) = \frac{\partial \log \mu(a, x; \beta)}{\partial \beta}.$$

Since $\widehat{\beta}$ is the maximum likelihood estimator of the true parameter $\beta^*$ based on $\{(a_i, x_i)\}_{i=1}^n$, we have

$$\frac{1}{n} \sum_{i=1}^n S(a_i, x_i; \widehat{\beta}) = 0. \tag{C.1}$$

On the other hand, we expand the above sum to obtain

$$\frac{1}{n} \sum_{i=1}^n S(a_i, x_i; \widehat{\beta}) = \frac{1}{n} \sum_{i=1}^n S(a, x; \beta^*) + \frac{1}{n} \sum_{i=1}^n \frac{\partial S(a, x; \beta^*)}{\partial \beta} \cdot (\widehat{\beta} - \beta^*)$$

$$+ \frac{1}{2n} \sum_{i=1}^n \frac{\partial^2 S(a, x; \beta_S)}{\partial \beta^2} (\widehat{\beta} - \beta^*, \widehat{\beta} - \beta^*, \cdot), \tag{C.2}$$

where $\beta_S = \lambda_S \beta^* + (1 - \lambda_S) \widehat{\beta}$ for some $\lambda_S \in [0, 1]$. Then by the definition of $\Delta_S$ in (A.2), we combine (C.1) and (C.2) to obtain

$$0 = \frac{1}{n} \sum_{i=1}^n S(a_i, x_i; \beta^*) + \mathbb{E}\left[\frac{\partial S(a, x; \beta^*)}{\partial \beta}\right] \cdot (\widehat{\beta} - \beta^*) - \Delta_S(\widehat{\beta} - \beta^*)$$

$$+ \frac{1}{2n} \sum_{i=1}^n \frac{\partial^2 S(a_i, x_i; \beta_S)}{\partial \beta^2} (\widehat{\beta} - \beta^*, \widehat{\beta} - \beta^*, \cdot). \tag{C.3}$$

Since we have

$$\mathbb{E}\left[\frac{\partial S(a, x; \beta^*)}{\partial \beta}\right] = -I(\beta^*),$$

which is invertible by Assumption 3.3, by rearranging the terms in (C.3) we obtain $\widehat{\beta} - \beta^*$

$$= I^{-1}(\beta^*) \cdot \left[\frac{1}{n} \sum_{i=1}^n S(a_i, x_i; \beta^*) - \Delta_S(\widehat{\beta} - \beta^*) + \frac{1}{2n} \sum_{i=1}^n \frac{\partial^2 S(a_i, x_i; \beta_S)}{\partial \beta^2} (\widehat{\beta} - \beta^*, \widehat{\beta} - \beta^*, \cdot)\right],$$

which concludes the proof of Lemma A.2. $\qquad\square$

## D    APPLICATION TO MULTINOMIAL LOGISTIC REGRESSION

In this section, we consider the setting where the logging distribution $\mu(a \,|\, x; \beta)$ is parametrized by the multinomial logistic regression model. We assume that $x \in \mathbb{R}^p$ and $a \in [m]$. Let

$$\beta = (\beta_1^\top, \ldots, \beta_{m-1}^\top, 0^\top)^\top, \quad \text{where} \ \ \beta_1, \ldots, \beta_{m-1} \in \mathbb{R}^p.$$

Thus, the dimension of the parameter $\beta$ is $d = p(m-1)$. The logging distribution $\mu(a \,|\, x; \beta)$ is parametrized by

$$\mu(a \,|\, x; \beta) = \frac{\exp(x^\top \beta_a)}{\sum_{l=1}^{m-1} \exp(x^\top \beta_l) + 1}. \tag{D.1}$$

We denote by $\lambda_{\max}(\cdot)$ and $\lambda_{\min}(\cdot)$ the largest and the smallest eigenvalue of a matrix, respectively. Throughout this section, we assume that the following regularity condition holds.

**Assumption D.1.** We assume that, for all $i \in [m-1]$, $\beta_i$ is bounded by $\|\beta_i\| \leq B$. Also, we assume that $r(a, x)$ is bounded for all $a$ and $x$. Furthermore, recall that in (2.1) we have $x \sim \nu$, we assume that $x$ is bounded by $\|x\| \leq X$ and for its covariance matrix $\Sigma = \mathbb{E}_{x \sim \nu}[xx^\top]$, we have

$$\lambda_{\max}(\Sigma) = \bar{\sigma}, \quad \lambda_{\min}(\Sigma) = \underline{\sigma} > 0. \tag{D.2}$$

Then the score function $S(a, x; \beta) \in \mathbb{R}^{p(m-1)}$ takes the form

$$S(a, x; \beta) = \begin{pmatrix} \dfrac{\exp(x^\top \beta_1)}{\sum_{l=1}^{m-1} \exp(x^\top \beta_l) + 1} \cdot x - \mathbb{1}(a = 1) \cdot x \\ \dfrac{\exp(x^\top \beta_2)}{\sum_{l=1}^{m-1} \exp(x^\top \beta_l) + 1} \cdot x - \mathbb{1}(a = 2) \cdot x \\ \vdots \\ \dfrac{\exp(x^\top \beta_{m-1})}{\sum_{l=1}^{m-1} \exp(x^\top \beta_l) + 1} \cdot x - \mathbb{1}(a = m - 1) \cdot x \end{pmatrix}, \qquad (D.3)$$

where $\mathbb{1}(\cdot)$ is the indicator function, while the Fisher information matrix $I(\beta) \in \mathbb{R}^{p(m-1) \times p(m-1)}$ is a block matrix whose $(j, k)$-th block, namely $[I(\beta)]_{j,k}$, takes the form

$$[I(\beta)]_{j,k} = \mathbb{E}\left[ \left( \mathbb{1}(j = k) - \frac{\exp(x^\top \beta_j)}{\sum_{l=1}^{m-1} \exp(x^\top \beta_l) + 1} \right) \cdot \frac{\exp(x^\top \beta_k)}{\sum_{l=1}^{m-1} \exp(x^\top \beta_l) + 1} \cdot xx^\top \right], \quad (D.4)$$

for $j, k \in [m - 1]$. We give lower and upper bounds for eigenvalues of $I(\beta^*)$ in the following lemma.

**Lemma D.2** (Non-singular Fisher Information Matrix). Under Assumption D.1, the Fisher information matrix $I(\beta)$ is non-singular. Furthermore, for the eigenvalues of $I(\beta)$, we have

$$\lambda_{\min}(I(\beta)) \geq \frac{2e^{-4XB}\underline{\sigma}}{m^2}, \quad \lambda_{\max}(I(\beta)) \leq \frac{4e^{4XB}\bar{\sigma}}{m^2}. \qquad (D.5)$$

*Proof.* For the ease of notation, we let $\pi_i = \mu(i \,|\, x, \beta)$ for $i \in [m - 1]$, $\pi_m = 1 - \sum_{i=1}^{m-1} \pi_i$ and $v = (v_1, \ldots, v_{m-1})^\top$ with $v_i \in \mathbb{R}^p$. First, for $x$ and $\beta_i$, $i \in [m - 1]$ satisfying Assumption D.1, we have

$$\pi_i \geq \frac{e^{-XB}}{(m - 1)e^{XB} + 1} \geq \frac{e^{-2XB}}{m}, \quad \pi_i \leq \frac{e^{XB}}{(m - 1)e^{-XB} + 1} \leq \frac{e^{2XB}}{m}, \qquad (D.6)$$

Then, by (D.4), we have

$$v^\top I(\beta)v = \sum_{1 \leq j,k \leq m-1} v_j^\top [I(\beta)]_{j,k} v_k$$

$$= \mathbb{E}\left[ \sum_{j=1}^{m-1} \pi_j (v_j^\top x)^2 + \sum_{j=1}^{m-1} \pi_j^2 (v_j^\top x)^2 - \sum_{1 \leq j,k \leq m-1} (\pi_j v_j^\top x)(\pi_k v_k^\top x) \right]. \qquad (D.7)$$

Note that $\sum_{k=1}^{m} \pi_k = 1$, we have

$$\sum_{j=1}^{m-1} \pi_j (v_j^\top x)^2 - \sum_{1 \leq j,k \leq m-1} (\pi_j v_j^\top x)(\pi_k v_k^\top x)$$

$$= \sum_{1 \leq j,k \leq m-1} \pi_j \pi_k (v_j^\top x)^2 + \pi_m \sum_{j=1}^{m-1} \pi_j (v_j^\top x)^2 - \sum_{1 \leq j,k \leq m-1} (\pi_j v_j^\top x)(\pi_k v_k^\top x)$$

$$= \frac{1}{2} \sum_{1 \leq j,k \leq m-1} (\pi_j v_j^\top x - \pi_k v_k^\top x)^2 + \pi_m \sum_{j=1}^{m-1} \pi_j (v_j^\top x)^2. \qquad (D.8)$$

Thus, we have

$$\lambda_{\min}(I(\beta)) = \min_{\|v\|=1} v^\top I(\beta)v$$

$$= \min_{\|v\|=1} \mathbb{E}\left[ \frac{1}{2} \sum_{1 \leq j,k \leq m-1} (\pi_j v_j^\top x - \pi_k v_k^\top x)^2 + \pi_m \sum_{j=1}^{m-1} \pi_j (v_j^\top x)^2 + \sum_{j=1}^{m-1} \pi_j^2 (v_j^\top x)^2 \right]$$

$$\geq \frac{2e^{-4XB}}{m^2} \min_{\|v\|=1} \mathbb{E}\left[ \sum_{j=1}^{m-1} (v_j^\top x)^2 \right] = \frac{2e^{-4XB}}{m^2} \lambda_{\min}(\Sigma) = \frac{2e^{-4XB}\underline{\sigma}}{m^2}, \qquad (D.9)$$

where the last two equalities are consequences of (D.6) and the Assumption D.1, respectively. This is saying that the Fisher information matrix $I(\beta)$ is positive definite. For the largest eigenvalue of $I(\beta)$,

we have

$$\lambda_{\max}\big(I(\beta)\big) = \max_{\|v\|=1} \mathbb{E}\left[\frac{1}{2} \sum_{1 \leq j,k \leq m-1} (\pi_j v_j^\top x - \pi_k v_k^\top x)^2 + \pi_m \sum_{j=1}^{m-1} \pi_j (v_j^\top x)^2 + \sum_{j=1}^{m-1} \pi_j^2 (v_j^\top x)^2\right]$$

$$\leq \frac{4e^{4XB}}{m^2} \max_{\|v\|=1} \mathbb{E}\left[\sum_{j=1}^{m-1} (v_j^\top x)^2\right] = \frac{4e^{4XB}\bar{\sigma}}{m^2},$$

which, together with (D.9), concludes the proof. $\qquad\square$

Meanwhile, the deviation function $D_V(r, a, x; \beta)$ takes the form

$$D_V(r, a, x; \beta) = \frac{\pi(a\,|\,x) \cdot r(a, x) \cdot \big(\sum_{l=1}^{m-1} \exp(x^\top \beta_l) + 1\big)}{\exp(x^\top \beta_a)} - V. \tag{D.10}$$

The gradient of the deviation function takes the form

$$\frac{\partial D_V(r, a, x; \beta)}{\partial \beta} = \pi(a\,|\,x) \cdot r(a, x) \cdot \begin{pmatrix} \dfrac{\exp(x^\top \beta_1) - \mathbb{1}(a = 1) \cdot \big(\sum_l \exp(x^\top \beta_l) + 1\big)}{\exp(x^\top \beta_a)} \cdot x \\ \dfrac{\exp(x^\top \beta_2) - \mathbb{1}(a = 2) \cdot \big(\sum_l \exp(x^\top \beta_l) + 1\big)}{\exp(x^\top \beta_a)} \cdot x \\ \vdots \\ \dfrac{\exp(x^\top \beta_{m-1}) - \mathbb{1}(a = m - 1) \cdot \big(\sum_l \exp(x^\top \beta_l) + 1\big)}{\exp(x^\top \beta_a)} \cdot x \end{pmatrix}. \tag{D.11}$$

The Hessian of the deviation function is a block matrix in $\mathbb{R}^{p(m-1)\times p(m-1)}$, whose $(j, k)$-th block takes the form

$$\left[\frac{\partial^2 D_V(r, a, x; \beta)}{\partial \beta^2}\right]_{j,k} = xx^\top \cdot \pi(a\,|\,x) \cdot r(a, x) \tag{D.12}$$

$$\cdot \left[\frac{\big(\mathbb{1}(k = j) - \mathbb{1}(k = a)\big)}{\exp\big(x^\top(\beta_a - \beta_j)\big)} + \frac{\mathbb{1}(k = a = j) \cdot \big(\sum_l \exp(x^\top \beta_l) + 1\big) - \mathbb{1}(a = j) \cdot \exp(x^\top \beta_k)}{\exp(x^\top \beta_a)}\right]$$

for $j, k \in [m - 1]$.

Note that by (D.3)-(D.4) and (D.11)-(D.12) for the population quantities, which do not scale with $n$, in Assumptions 3.3-3.4 are all bounded under Assumption D.1. Hence, in the multinomial logistic regression model defined in (D.1) with bounded $x$ and $\beta$ as stated in Assumption D.1, when $d$ is fixed, we have

$$\zeta_{\mathrm{DG}} = \zeta_{\mathrm{DH}} = \zeta_{\mathrm{ST}} = 1, \quad \text{and } t \leq M, \tag{D.13}$$

where $t$ is defined in Condition 3.12 and $M$ is some constant that does not scale with $n$.

The subsequent lemmas establish the tail behaviors for the events defined in Definitions 3.5-3.7, which together yield a concrete form of the tail function $f(t)$ defined in Condition 3.8.

**Lemma D.3** (Maximum Likelihood Estimation Error). Under the Assumption D.1, for the event $\mathcal{E}_\beta(t_\beta)$ defined in (3.8), we have

$$\mathbb{P}\big(\mathcal{E}_\beta(t_\beta)\big) \geq 1 - C_\beta \cdot \exp(d - nt_\beta^2),$$

where $C_\beta$ is a positive constant independent of $n$.

*Proof.* Let $\widehat{\ell}_n(\beta)$ be the negative log-likelihood function $\widehat{\ell}_n(\beta) = -(1/n) \cdot \sum_{i=1}^n \log \mu(a_i, x_i; \beta)$, where $\mu(a, x; \beta)$ is defined in (D.1). By Lemma D.2, we have $\lambda_{\min} \geq 2e^{-4XB}\underline{\sigma}/m^2$, which gives

$$
\mathbb{P}\left(\lambda_{\min}\left(\nabla^2\widehat{\ell}_n(\beta^*)\right) \geq \frac{e^{-4XB}\underline{\sigma}}{m^2}\right) \geq \mathbb{P}\left(\lambda_{\min}\left(\nabla^2\widehat{\ell}_n(\beta^*)\right) \geq \left(1 - \frac{t_\beta}{2M}\right) \cdot \frac{2e^{-4XB}\underline{\sigma}}{m^2}\right)
$$

$$
\geq \mathbb{P}\left(\left|\lambda_{\min}\left(\nabla^2\widehat{\ell}_n(\beta^*)\right) - \lambda_{\min}\left(I(\beta^*)\right)\right| \leq \frac{e^{-4XB}\underline{\sigma}}{Mm^2} \cdot t_\beta\right)
$$

$$
\geq \mathbb{P}\left(\left\|\nabla^2\widehat{\ell}_n(\beta^*) - I(\beta^*)\right\| \leq \frac{e^{-4XB}\underline{\sigma}}{Mm^2} \cdot t_\beta\right), \qquad (D.14)
$$

where $M$ is defined in (D.13). Then note that $\left\|\partial^2 \log \mu(a_i, x_i; \beta^*)/\partial\beta^2 - I(\beta^*)\right\| \leq 8e^{4XB}\bar{\sigma}/m^2$ and $\|I(\beta^*)^2\| \leq X^2 e^{4XB}\bar{\sigma}m$, by matrix Bernstein inequality (see, for example, Tropp (2012); Vershynin (2010)), we have

$$
\mathbb{P}\left(\left\|\nabla^2\widehat{\ell}_n(\beta^*) - I(\beta^*)\right\| \leq \frac{e^{-4XB}\underline{\sigma}}{Mm^2} \cdot t_\beta\right) \leq d \cdot \exp\left(\frac{-nt_\beta^2}{(6X^2 e^{4XB}Mm^3 + 16\underline{\sigma}t_\beta)M\bar{\sigma}m^2/3\underline{\sigma}^2}\right)
$$

$$
\leq C'_\beta \exp(d - nt_\beta^2), \qquad (D.15)
$$

where the second inequality holds for a constant $C'_\beta$ when $n$ sufficiently large. Combining (D.14) and (D.15), we know that $\widehat{\ell}_n(\beta)$ is $e^{-4XB}\underline{\sigma}/m^2$-strongly convex with at least probability $1 - C'_\beta \exp(d - nt_\beta^2)$. The rest of the proof considers the case where the $e^{-4XB}\underline{\sigma}/m^2$-strong convexity holds for $\widehat{\ell}_n(\beta)$, which means that we have

$$
\widehat{\ell}_n(\widehat{\beta}) \geq \widehat{\ell}_n(\beta^*) + \nabla_\beta\widehat{\ell}_n(\beta^*)^\top(\widehat{\beta} - \beta^*) + \frac{e^{-4XB}\underline{\sigma}}{2m^2} \cdot \|\widehat{\beta} - \beta^*\|^2. \qquad (D.16)
$$

By definition of MLE, $\widehat{\beta}$ is the minimizer for $\widehat{\ell}_n(\beta)$, which means $\widehat{\ell}_n(\beta^*) \geq \widehat{\ell}_n(\widehat{\beta})$. Therefore, by Cauchy-Schwarz inequality, we have

$$
\frac{e^{-4XB}\underline{\sigma}}{2m^2} \cdot \|\widehat{\beta} - \beta^*\|^2 \leq \nabla_\beta\widehat{\ell}_n(\beta^*)^\top(\beta^* - \widehat{\beta}) \leq \|\nabla_\beta\widehat{\ell}_n(\beta^*)\| \cdot \|\widehat{\beta} - \beta^*\|,
$$

which gives

$$
\mathbb{P}\left(\|\widehat{\beta} - \beta^*\| \geq t_\beta\right) \leq \mathbb{P}\left(\|\nabla_\beta\widehat{\ell}_n(\beta^*)\| \geq \frac{e^{-4XB}\underline{\sigma}}{2m^2} \cdot t_\beta\right). \qquad (D.17)
$$

Note that $\mathbb{E}[S(a, x; \beta^*)] = 0$, $\mathbb{E}[S(a, x; \beta^*)S(a, x; \beta^*)^\top] = I(\beta^*)$ and $\nabla_\beta\widehat{\ell}_n(\beta^*) = (1/n) \cdot \sum_{i=1}^n S(a_i, x_i; \beta^*)$, by Bernstein's inequality (see, for example, Tropp (2012)), we have

$$
\mathbb{P}\left(w^\top\nabla_\beta\widehat{\ell}_n(\beta^*) \geq \frac{e^{-4XB}\underline{\sigma}}{2m^2} \cdot t_\beta\right)
$$

$$
\leq \exp\left(\frac{-nt_\beta^2}{[8e^{8XB}\bar{\sigma}/m^2 + 2X(1 + e^{2XB})e^{4XB}t_\beta/3]e^{4XB}\underline{\sigma}^{-2}m^2}\right) \leq C''_\beta \exp(-nt_\beta^2), \quad (D.18)
$$

for any $w$ such that $\|w\| = 1$, where the second inequality holds for a constant $C''_\beta$ when $n$ is large enough. Let $\mathcal{G}_d = \{w \in \mathbb{R}^d : \|w\| = 1\}$, by taking a union bound over $w \in \mathcal{G}_d$ in (D.18), we have

$$
\mathbb{P}\left(\|\nabla_\beta\widehat{\ell}_n(\beta^*)\| \geq \frac{e^{-4XB}\underline{\sigma}}{2m^2} \cdot t_\beta\right) = \mathbb{P}\left(\sup_{w \in \mathcal{G}_d} w^\top\nabla_\beta\widehat{\ell}_n(\beta^*) \geq t_\beta\right) \leq C''_\beta \exp(d - nt_\beta^2). \quad (D.19)
$$

Plugging (D.19) into (D.17) and recall that (D.16) holds with at least a probability of $1 - C'_\beta \exp(d - nt_\beta^2)$, we have

$$
\mathbb{P}(\mathcal{E}_\beta(t_\beta)) \leq \left(1 - C'_\beta \exp(d - nt_\beta^2)\right) \cdot \left(1 - C''_\beta \exp(d - nt_\beta^2)\right) \leq 1 - (C'_\beta + C''_\beta)\exp(d - nt_\beta^2),
$$

which concludes the proof with $C_\beta = C'_\beta + C''_\beta$. $\qquad \square$

Next, we state two Lemmas on the concentration of score and deviation. The proofs for them are quite standard in matrix concentration theories.

**Lemma D.4** (Concentration of Score). Under the Assumption D.1, for the logistic regression model in (D.1), with the event $\mathcal{E}_{\mathrm{SG}}(t_{\mathrm{SG}})$, $\mathcal{E}_{\mathrm{SH}}(t_{\mathrm{SH}})$ and $\mathcal{E}_{\mathrm{ST}}(t_{\mathrm{ST}})$ defined in Definition 3.6, we have

$$\mathbb{P}\big(\mathcal{E}_{\mathrm{SG}}(t_{\mathrm{SG}})\big) \geq 1 - C_{\mathrm{SG}} \cdot \exp(d - nt_{\mathrm{SG}}^2), \quad \mathbb{P}\big(\mathcal{E}_{\mathrm{SH}}(t_{\mathrm{SH}})\big) \geq 1 - C_{\mathrm{SH}} \cdot \exp(d - nt_{\mathrm{SH}}^2)$$

and

$$\mathbb{P}\big(\mathcal{E}_{\mathrm{ST}}(t_{\mathrm{ST}})\big) \geq 1 - C_{\mathrm{SG}} \cdot \exp(d - nt_{\mathrm{ST}}^2),$$

where $C_{\mathrm{SG}}$, $C_{\mathrm{SH}}$, and $C_{\mathrm{ST}}$ are positive constants independent of $n$.

*Proof.* Following the arguments made in (D.15) and (D.19), the exponentially decaying tails are immediate. $\qquad \square$

**Lemma D.5** (Concentration of Deviation). Suppose the Assumption D.1 holds and $r(a, x) \leq R$ for all $a$ and $x$. Then, for the logistic regression model (D.1), with the event $\mathcal{E}_{\mathrm{DG}}(t_{\mathrm{DG}})$ and $\mathcal{E}_{\mathrm{DH}}(t_{\mathrm{DH}})$ defined in 3.7, we have

$$\mathbb{P}\big(\mathcal{E}_{\mathrm{DG}}(t_{\mathrm{DG}})\big) \geq 1 - C_{\mathrm{DG}} \cdot \exp(d - nt_{\mathrm{DG}}^2), \quad \mathbb{P}\big(\mathcal{E}_{\mathrm{DH}}(t_{\mathrm{DH}})\big) \geq 1 - C_{\mathrm{DH}} \cdot \exp(d - nt_{\mathrm{DH}}^2), \tag{D.20}$$

where $C_{\mathrm{DG}}$ and $C_{\mathrm{DH}}$ are positive constants independent of $n$.

*Proof.* Recall that we have $\pi_i \geq e^{-2XB}/m$ when the Assumption D.1 holds. We have

$$\left\| \frac{\partial D_V(r, a, x; \beta)}{\partial \beta} \right\| \leq RX\big[(m-2)e^{2XB} + me^{2XB}\big] = 2e^{2XB}RX(m-1),$$

and, by similar arguments in the proof of Lemma D.2, we have

$$\left\| \mathbb{E}\left[ \frac{\partial D_V(r, a, x; \beta)}{\partial \beta} \cdot \frac{\partial D_V(r, a, x; \beta)}{\partial \beta^\top} \right] \right\| \leq R^2 \left( \frac{m}{e^{-2XB}} \right)^2 \bar{\sigma} = e^{4XB}R^2\bar{\sigma}m^2.$$

Then, the first inequality in (D.20) follows by similar arguments as in (D.17)-(D.18). Next, again by similar arguments in the proof of Lemma D.2, we also have

$$\left\| \mathbb{E}\left[ \frac{\partial^2 D_V(r, a, x; \beta)}{\partial \beta^2} \right] \right\| \leq e^{2XB}R\bar{\sigma}(m+1)$$

and

$$\left\| \mathbb{E}\left[ \left( \frac{\partial^2 D_V(r, a, x; \beta)}{\partial \beta^2} \right)^2 \right] \right\| \leq e^{2XB}R^2X^2\bar{\sigma}m(m+1),$$

Then, similar as (D.15), the second inequality in (D.20) follows by matrix Bernstein inequality. $\qquad \square$

Recall the event $\mathcal{E}(t^\star)$ with $t^\star = \|t\|_\infty$ is defined in Condition 3.8. By Lemmas D.3-D.5, the tail function $f(t)$ takes the form

$$f(t) \leq 6C \cdot \exp(d - nt^2), \tag{D.21}$$

where $C = \max\{C_\beta, C_{\mathrm{SG}}, C_{\mathrm{SH}}, C_{\mathrm{DG}}, C_{\mathrm{DH}}, C_{\mathrm{DT}}\} > 0$.

In the sequel, we use (D.13) and the tail function in (D.21) to characterize the reduction of the mean squared error induced by the multinomial logistic regression model.

First, let $t_0 = (T+1)\sqrt{d/n}$ with some constant $T > 0$, we have the following bound on the tail probability of $t$ based on (D.21),

$$\mathbb{P}(t > t_0) \leq f(t_0) \leq 6C \cdot e^{-Td}. \tag{D.22}$$

Then recall that in Theorem 3.9 we have

$$\xi(n) = \left( \mathbb{E}\big[(\widetilde{V} - V)^2\big] \right)^{1/2} \cdot O\left( \left( \int_0^\infty t^4 \,\mathrm{d}f(t) \right)^{1/2} \right) + O\left( \int_0^\infty t^3 \,\mathrm{d}f(t) \right),$$

where by (D.22), we have

$$\int_0^{t_0} t^3 \,\mathrm{d}f(t) = \mathbb{E}[t^3 \,|\, t \leq t_0] \leq (T+1)^3 \cdot (d/n)^{3/2} = O\big((d/n)^{3/2}\big), \tag{D.23}$$

and

$$\int_{t_0}^{M} t^3 \, \mathrm{d}f(t) = \mathbb{E}[t^3 \,|\, t > t_0] \leq \mathbb{P}(t > t_0) \cdot M^3 \leq 6C \cdot \exp(-Td) \cdot M^3 = O(e^{-Td}), \quad \text{(D.24)}$$

which decays exponentially with $d$ and is therefore dominated by the bound given in (D.23). Hence, we have

$$\int_0^{\infty} t^3 \, \mathrm{d}f(t) = \int_0^{t_0} t^3 \, \mathrm{d}f(t) + \int_{t_0}^{M} t^3 \, \mathrm{d}f(t) = O\big((d/n)^{3/2}\big). \quad \text{(D.25)}$$

Using the same arguments in derivation of (D.25), we also have

$$\left( \int_0^{\infty} t^4 \, \mathrm{d}f(t) \right)^{1/2} = O\big(d/n\big).$$

Hence, we obtain

$$\xi(n) = \left( \mathbb{E}\big[(\widetilde{V} - V)^2\big] \right)^{1/2} \cdot O\big(d/n\big) + O\big((d/n)^{3/2}\big).$$

Note that the mean squared error term $\mathbb{E}[(\widetilde{V} - V)^2]$ is of the order $1/n$, which implies that $\xi(n)$ is of the order $(d/n)^{3/2}$. Thus, (3.15) is satisfied when $n$ is sufficiently large, which means that a reduction of the order $1/n$ on the mean squared error is achieved by the multinomial logistic regression model when the sample size $n$ is sufficiently large. This result implies that the MLIPS in (3.1) is guaranteed to out perform the IPS estimator in (2.2) when the sample size $n$ is much larger than the dimension $d$ of the problem. Moreover, since the mean squared error term $\mathbb{E}[(\widetilde{V} - V)^2]$ itself is of the order $1/n$, the effect of reduction does not vanish when $n \to \infty$.

We interpret Theorem 3.9 in a more general way. The $\xi(n)$ term is in general a decreasing function of the sample size $n$ and is mostly of a higher order than $1/n$ in common parametric statistical models. Correspondingly, for a sufficiently large $n$, we obtain (3.15), which implies that our method leads to a $1/(2n) \cdot \mathrm{Var}(\Pi(r, x, a; \beta^*))$ reduction on the mean squared error, which has a non-vanishing effect when $n \to \infty$.

# E COMPLIMENTARY RESULTS FOR SECTION 4.3

We provide the following extra results in Table 2 for the experiment in Section 4.3. The additional results of standard deviation of the Hamming losses of the policies optimized by using IPS, POEM and their ML-variants are computed over ten runs. From the Table 2, we can see that our ML-based approach induces much smaller standard deviations than the improvements made on performances and, thus, it consistently boosts the performance of IPS and POEM in policy optimization.

Table 2: Hamming losses and standard deviations ('Std' columns in the table) for the performances (evaluated by Hamming loss) of IPS, POEM and their ML-variants on the datasets of Swaminathan & Joachims (2015a). Here, $10^{-k}$ is denoted by e-$k$.

| Dataset/Alg. | Scene | (Std) | Yeast | (Std) | TMC | (Std) | LYRL | (Std) |
|---|---|---|---|---|---|---|---|---|
| IPS | 1.342 | (3.14e-1) | 4.571 | (1.40e-1) | 3.023 | (N/A) | 1.108 | (9.18e-2) |
| POEM | 1.143 | (2.91e-4) | 4.549 | (1.05e-1) | 2.522 | (N/A) | 0.981 | (3.31e-2) |
| MLIPS-Lin | 1.086 | (6.26e-6) | 3.778 | (9.98e-2) | 2.018 | (6.23e-3) | 1.025 | (5.92e-2) |
| MLIPS-NN | 1.086 | (4.88e-6) | 3.630 | (9.10e-2) | 2.019 | (1.22e-3) | 0.930 | (2.08e-2) |
| MLPOEM-Lin | 1.086 | (3.48e-6) | 3.894 | (3.16e-1) | 2.010 | (7.96e-3) | 0.949 | (4.21e-2) |
| MLPOEM-NN | 1.086 | (5.81e-6) | 3.477 | (2.03e-1) | 2.000 | (1.22e-2) | 0.904 | (2.46e-2) |

