# OpenReview forum: "Off-Policy Evaluation and Learning from Logged Bandit Feedback: Error Reduction via Surrogate Policy"
_ICLR.cc/2019/Conference_

### Official Review · AnonReviewer3 · 2018-11-01
**An interesting idea with promising empirical results and a somewhat disappointing theoretical analysis.**

**Rating:** 6
**Confidence:** 3

**Review:**

This work is concerned with the problem of batch contextual bandits, in which a target contextual bandit policy is optimized on the data generated by a different logging policy. The main problem is to come up with a low-variance low-bias estimator for the value of the target policy. Many of the known techniques are based on an unbiased estimator known as inverse propensity scoring (IPS), which uses the distribution over actions of the logging policy, conditioned on the observed contexts. However, IPS suffers from large variance. The paper's idea is to do a maximum likelihood fit of a simple surrogate policy to the logged data, and then use the conditional distribution over actions of the surrogate policy to compute inverse propensity scores.
The theoretical results show that the bias of this estimator vanishes asymptotically, whereas the variance is smaller than IPS. Experiments using known/unknown logging policies on artificial/real-world bandit data show that the IPS scores computed with the proposed technique are empirically better than those computed directly using the logging policy. Moreover, the advantage increases when the distribution extracted from the surrogate policy is used to compute more sophisticated estimators than IPS.

The off-policy evaluation in contextual bandits is an important problem, and this paper appears to make some progress. However, the theoretical analysis is a bit disappointing, as it does not shed much light on the reasons why using a surrogate policy should help. Some additional discussion would add value to the paper.

The result about the decrease in variance depends on assumptions that are not clearly justified, and is expressed in terms of abstract quantities that hard to connect to concrete scenarios. In the end, one does not get many new insights from the theory.

In Assumptions 3.3-3-4, what is the variable w.r.t the asymptotic notations are understood? By that I mean, the variable n such that f(n) = O(g(n)).

The experiments are competent and quite elaborated. However, the statistical significance of the improvements in Table 1 is unclear.

The evaluation criterion for the Criteo experiment is unclear. As a consequence it is hard to appreciate the significance of the improvements in this case.

---

> ### Author Response · Authors · 2018-11-20
> **Thanks for you comments. We have revised accordingly and added extra experiment results.**
>
> Thanks for your valuable comments. In the following we address the issues raised in the comments. Please find the corresponding revisions in our updated paper.
>
> Insight of Theoretical Analysis: In the revised version, we have highlighted the insight behind the reduction of MSE after stating the main theorem.
> Here is a brief explanation of the intuition behind the analysis: The reason behind the reduction of MSE is due to the Eq(A.16) we derived in the proof of Theorem 3.9. Interpretation for the equation is that $(\tilde{V} - V)^2$ and $\Pi^2(r, x, a; \beta^*)$ are orthogonal in expectation. Therefore, we have $(1/n)Var(\Pi(r, x, a; \beta^*))$ as the MSE reduction term. For the rest parts of the proof, as we perform a non-asymptotic analysis in this paper, many efforts are taken to present the concentration condition and bound the residual term $\xi(n)$ to ensure that it doesn’t dominate the MSE reduction term.
>
> The Assumptions: In the paper, we try to make our assumption as mild as possible to fit in more general cases. As a consequence, they become more abstract to be able to fit in more general cases. To make the presentation clearer, we will add more explanations and examples for the assumptions in the revision.
> 1)  We allow flexibility in these the Assumptions 3.3-3.4, i.e., those O(\zeta)’s are not specified, which means they can depend on what kind of logging distribution $\mu$ we are handling with. It is worth noting that as these two assumptions are made on population quantities, which do not scale with n, they do not affect our main theorem provided that the sample size n is reasonably large.
> 2) Our intuition for the Condition 3.8 is that when the log data are i.i.d. samples, the Condition 3.8 can be verified by using Bernstein-type concentration inequalities. Those inequalities have exponentially decaying tails, which are stronger than the super polynomial decaying requirement made in Condition 3.8.
>
> Table 1: We have added some extra results on the four datasets in Table 1. These extra results are the standard deviation of the performances of policy optimization (not evaluation, which we already discussed in Section 4.2) by IPS & MLIPS and POEM & MLPOEM. From the table, we can see that our ML-based approach has much smaller standard deviations than the boosts in performances. So our approach boosts the performance of IPS and POEM significantly in policy optimization with high probability. For details, please check the Appendix E in the updated paper.
>
> Asymptotic Notions: For a population quantity $f$, the asymptotic notion $f = O(\zeta)$ in Assumptions 3.3-3.4 mean that $f(d, model) \leq C\zeta(d, model)$, where $C$ is some positive constant that depend only on the dimension d and necessary regularity conditions for the model of logging distribution.
>
> Criteo Experiment: The training dataset was given while the test dataset was held out by the challenge organizer. During that challenge, we apply policy optimization over the training set, then give our optimized policy back to them. Then they evaluate the performance of our policy on the hold out testing dataset. The values we include in our paper are the rewards of the policies (the higher the better). The improvements made us rank among prize wining teams in that challenge, which convince us that the ML-based approach did achieve significant improvements in this case.

---

### Official Review · AnonReviewer2 · 2018-11-02
**A variance reduction technique for learning from logged bandit feedback. Proposes a surrogate policy that can be used on top of IPS and POEM.**

**Rating:** 8
**Confidence:** 4

**Review:**

Summary:
The paper considers the problem of learning from logged bandit feedback, and focuses on the problem of the ratio of the target policy and the logged policy (the basis of algorithms such as inverse propensity scoring). The paper proposes a surrogate policy to replace the logged policy with known parametrization, with a policy obtained by maximum likelihood estimation on the observed data. The authors present theoretical arguments that the variance of the value function estimate is reduced. Empirical experiments show that the surrogate policy can be used to improve IPS and POEM, and also works when the logging policy is unknown.

The paper analyses an important and interesting problem which is critical to many practical applications today. The proposed solution is modular, and the empirical experiments point to its usefulness. The theoretical analysis, while not fully explaining the proposed approach, provides comfort that there is reduced variance when using the maximum likelihood surrogate.

Overall comments:
- page 3, Section 3: It is unclear why the assumption that we know the logging policy, as well as its optimal parameter is a sensible one. In particular, the first paragraph seems to indicate that the surrogate policy some somehow the same parameterization and $\hat{\beta}$ is in the same space as $\beta^*$, and just a different parameter. On one hand the authors seem to indicate that they know everything about the logging. On the other hand they seem to want to claim that not knowing the logging policy is ok. What happens when there is a model mismatch between the logging policy and the surrogate policy? Please expand on these two assumptions.
- page 4, Section 3.1: It might be useful to have a toy example which exactly matches the requirements of Theorem 3.9, such that you can present empirical intuition about the terms in (3.13). In particular: what is the effect of assuming a deterministic reward? How does (3.14) grow? Why is the reduction of MSE greater than $\xi(n)$?
- Theorem 3.9: Please present the result that MLIPS is asympotically unbiased explicitly. Furthermore, the current proof of this main theorem should be structured better, so that it can be properly checked.

Minor issues/typos:
- page 3, above (3.1): In specific, we --> In particular, we
- Figure 1: the legend is very confusing, making it totally unclear what the text is talking about. Please match text, caption and legend.
- Section 4.3: please say that the data is the multilabel datasets of Swaminathan and Joachims in Table 1.

---

> ### Author Response · Authors · 2018-11-20
> **Thanks for you comments. We have revised accordingly.**
>
> Thank you for your valuable comments. In the following we address the issues raised in the comments. Please find the corresponding revisions in our updated paper.
>
> Section 3 (Knowledge of Logging Distribution): We have added some discussion over this issue in the updated paper.
> Throughout the theoretical analysis, we assume that the parametrization of the logging policy is known. Our theoretical results show that we can benefit from estimating the parameter within the parametrization family of distributions using MLE. Even when the true parameter $\beta^*$ is known beforehand, this approach reduces MSE of the policy estimation.
> As the reviewers have pointed out, the parametrization of the logging distribution may be misspecified in practice when we approximate using some model. However, when universal function approximators such as neural networks are used, the approximation error (bias) often diminishes with an increasing number of layers and neurons (see, for example, [1-3]). Such approximation error enters the Taylor expansions in Lemma A.1-A.2.
>
> Section 3.1:
> 1)(Deterministic Reward): In fact, approaches based on propensity scores are unable to handle cases where the reward has endogenous randomness, i.e., the randomness depends on x and a. It complicates our notation a bit by adding an extra expectation nation on the reward if we allow exogenous randomness to the reward, i.e., the randomness is independent of x and a, but this does not change the arguments we make.
> Thus, here we make the reward deterministic for the simplicity of analysis and for better understanding of the source of the improvement made by our ML-based approach, which is due to the use of MLE of logging distribution parameter in IPS.
>
> 2)(Toy Example): We use the multinomial logistic regression as an example. The detailed proof is deferred to Appendix D. At the end of Appendix D, we quantify with a MSE reduction of order O(1/n) for multinomial logistic regression, while the term $\xi(n)$ is of order $O((d/n)^{3/2})$, which is small compared to the reduction of MSE. Furthermore, as the MSE term itself is of order O(1/n) (the same order in n as the reduction term) when the samples are i.i.d., so we also illustrate that the MSE reduction does not vanish asymptotically for regression.
>
> Theorem 3.9:
> 1) (Asymptotic Unbiasedness): The asymptotic unbiasedness is proved in Proof of Theorem 3.9 in the Appendix A. Please see Eq(A.3)-(A.7).
> 2) (Structure of Proof): The proof is is decomposed to the following three parts. First, we introduce Lemma A.1 and A.2 to get Eq(A.3). Then, we analyze three terms in Eq(A.8) separately. Among the three terms, (i) is the most crucial one while the rest two terms are small terms compared to (i). Finally, we conclude our proof by applying the Condition 3.8 on tail behavior to Eq(A.26).
>
> Minor issues/typos: Thanks for pointing them out, we have revised accordingly in the updated paper.
>
> [1] Schmidt-Hieber, J., 2017. Nonparametric regression using deep neural networks with ReLU activation function. arXiv preprint arXiv:1708.06633.
> [2] Yarotsky, D., 2017. Error bounds for approximations with deep ReLU networks. Neural Networks, 94, pp.103-114.
> [3] Telgarsky, M., 2016. Benefits of depth in neural networks. arXiv preprint arXiv:1602.04485.

---

### Official Review · AnonReviewer1 · 2018-11-03
**An interesting approach to improve off-policy optimization in bandit settings by estimating the logging policy that generated the data**

**Rating:** 6
**Confidence:** 4

**Review:**

The paper proposes to fit a model of the logging policy that generates bandit feedback data, and use this model's propensities when performing off-policy optimization. When the model is well-specified (i.e. the logging policy indeed lies within the parametric class of models we are fitting), and we use maximum likelihood estimation to fit the model, this approach can yield a lower error when evaluating a policy's performance using off-policy data. The paper then shows how this improved off-policy estimation can also yield better off-policy optimization, and demonstrate this in semi-synthetic experiments.

Specific Comments:
Eq2.4: Lambda is overloaded (context distribution vs. regularization hyper-parameter).
Eq3.3: E[.] is used before defining it (i.e., E[.] should be interpreted as E_(x,a)~mu(.|beta*) [.])
Eq3.5: I^-1(beta*) makes sense, but the second term E[ d/d beta (S(x,a; beta*)) ] uses a notation that needs to be introduced (you mean || (E[ d/d beta (S(x,a; beta)) ] |_at beta=beta* )^-1 ||).

After Eq3.7: It will be instructive to specify some examples of logging policies mu which satisfy these assumptions (and how big the O(.) constants are for those examples).
Section 3.2: In practical considerations, expected a discussion of how robust things are when the logging policy class is mis-specified (i.e. assuming there is a beta* such that mu(.|beta*) created the data is unlikely to be true).
For ML- approaches, was a clipping constant M still used? If so, was it crucial and why?
Lemma D.1: The lemmas in appendix should be accompanied by a proof. E.g. what is C_beta? I don't immediately see why D.3 suggests that the inverse of the Fisher matrix has bounded norm (for instance, if x=0 the inverse is undefined).

General Comments:
Clarity: Good. The paper is easy to follow. Some examples from the Appendix can be moved to the main text (especially to provide a firm grounding for the constants appearing in Section3.1)
Correctness: I did not step through Appendix A-C. In Appendix D, there was a questionable claim. The stated theorems in the main text are believable [not surprising that asymptotic bias vanishes when the logging policy model is well-specified].
Originality: This builds on several previous works on off-policy optimization in bandit settings, and proposes a simple addition to improve performance.
Significance: The paper seems to have missed an opportunity; it can be substantially stronger with a more careful study of when fitting the logging policy will help vs. hurt, and what kinds of regularization or alternatives to maximum likelihood estimation can yield similar improvements (e.g. regularizing propensities close to uniform, ensure no small propensities).

---

> ### Author Response · Authors · 2018-11-20
> **Thanks for you comments. We have revised accordingly.**
>
> Thank you for your valuable comments. In the following we address the issues raised in the comments. Please find the corresponding revisions in our updated paper.
>
> Eq 2.4 and Eq 3.3: Thanks for pointing them out. We have revised accordingly.
>
> Eq 3.5: Thanks for pointing it out. We have made it clear what we mean by writing “d/d beta (S(x, a; beta*))” in the updated paper.
>
> After Eq 3.7: In the appendix of our updated paper, we have specified how these assumptions are satisfied by a logging policy that follows the multinomial logistic model.
>
> Model Misspecification: In practice, for logged bandit data, when the logging distribution is unavailable, we need to approximate it in order to at least calculate the propensity scores, i.e., $\pi(a | x)/\mu(a | x)$. There is a risk of model misspecification for those universal function approximators such as neural networks. However, the approximation error by neural network is usually very small and ,moreover, decreases with the number of layers and neurons, as shown by a number of recent works[1-3]. Such a diminishing approximation error should enter the Taylor expansions in Lemma A.1-A.2. We have included discussion on this problem in the Section 3.2 in the update paper.
>
> Clipping Constant: We do not introduce extra clipping constant in our ML-based approach. However, our approach is orthogonal and compatible with those importance weighted estimators with clipping constants. For example, as we illustrate in our experiments, the performance of the POEM algorithm can be boosted by ML-based approach. The POEM algorithm is based on the propensity weight capping approach, which has a clipping constant M.
>
> Lemma D.1: We have added a proof for the lemma in our update version, please see the proof Lemma D.3 in the updated paper.
>
> Eq(D.3): We have added a lemma with proof of the non-singularity as well as upper and lower bounds for eigenvalues of Fisher information after the equation under some regularity conditions. Please see Lemma D.2 in the updated paper.
>
> Alternative Techniques: In fact, the POEM algorithm is a technique using a clipping constant to ensure no small propensities. The variance reduction made by POEM can be further boosted by our ML-based approach as shown in the experiments in Section 4.3.
>
> [1] Schmidt-Hieber, J., 2017. Nonparametric regression using deep neural networks with ReLU activation function. arXiv preprint arXiv:1708.06633.
> [2] Yarotsky, D., 2017. Error bounds for approximations with deep ReLU networks. Neural Networks, 94, pp.103-114.
> [3] Telgarsky, M., 2016. Benefits of depth in neural networks. arXiv preprint arXiv:1602.04485.

---

### Author Response · Authors · 2018-11-26
**Revision: More practical discussion + detailed proofs for multinomial logistic regression model + extra experiment results**

We have made a revision of our paper. We included 5-6 pages of extra details and proofs. The major changes are summarized as follows:

(1). We highlighted a key fact (orthogonality between $\Pi$ and $\tilde{V} - V - \Pi$) for a better understanding of the main theorem on page 6, before the interpretation of the main theorem.

(2). We added some discussion on the possible model misspecification when fit the logging policy by function approximators.

(3). We added more technical details to Appendix D (Application to Multinomial Logistic Regression). In specific, based on Assumption D.1, we added Lemma D.2, along with a proof, for non-singularity of the Fisher information matrix, a proof of Lemma D.3 (which is Lemma D.1 in the previous version) and proof sketches of Lemma D.4-D.5 (which are Lemmas D.2 and D.3 in the previous version).

(4). We also included additional experiment results in Appendix E to address the statistical significance of our ML-based approach. Table 2 in Appendix E gives the standard deviation of performances of policy optimizations with different estimators (IPS, MLIPS, POEM and MLPOEM). The standard deviation of performances of ML-based techniques is much smaller than the improvements in all four bandit datasets, which means that our approach makes statistically significant improvements over the original methods.

Finally, we include the link (https://www.crowdai.org/challenges/nips-17-workshop-criteo-ad-placement-challenge) for the NIPS’17 Criteo Ad Placement Challenge in Section 4.3.1. We placed the 3rd in that challenge with a 54.314 IPS. It is worth noting that our IPS_std is also very small, which indicates a good statistical strength of our approach.

---

### Meta-Review · Area_Chair1 · 2018-12-17
**Interesting improvement to inverse propensity weighting based estimators for off-policy evaluation**

**Confidence:** 5
**Recommendation:** Accept (Poster)

**Metareview:**

This is an interesting paper that shows how improved off-policy estimation (and optimization) can be improved by explicitly estimating the data logging policy.  It is remarkable that the estimation variance can be reduced over using the original logging policy for IPW, although this result depends on the (somewhat impractical) assumption that the parametric form for the true logging policy is known.  The reviewers unanimously recommended the paper be accepted.  However, there remain criticisms of the theoretical analysis that the authors should take into account in preparing a final version (namely, motivating the assumptions needed to obtain the results, and providing stronger intuitions behind the reduced variance).